# Global population genomic signature of *Spodoptera frugiperda* (fall armyworm) supports complex introduction events across the Old World

Wee Tek Tay [1][✉], Rahul V. Rane [1], Amanda Padovan [1], Tom K. Walsh[1], Samia Elfekih[2], Sharon Downes[3], Kiwong Nam[4], Emmanuelle d'Alençon [4], Jianpeng Zhang[5], Yidong Wu [5], Nicolas Nègre [4], Daniele Kunz[6], Darren J. Kriticos [1], Cecilia Czepak[7], Michael H. Otim[8] & Karl H. J. Gordon[1]

Native to the Americas, the invasive *Spodoptera frugiperda* (fall armyworm; FAW) was reported in West Africa in 2016, followed by its chronological detection across the Old World and the hypothesis of an eastward Asia expansion. We explored population genomic signatures of American and Old World FAW and identified 12 maternal mitochondrial DNA genome lineages across the invasive range. 870 high-quality nuclear single nucleotide polymorphic DNA markers identified five distinct New World population clusters, broadly reflecting FAW native geographical ranges and the absence of host-plant preferences. We identified unique admixed Old World populations, and admixed and non-admixed Asian FAW individuals, all of which suggested multiple introductions underpinning the pest's global spread. Directional gene flow from the East into eastern Africa was also detected, in contrast to the west-to-east spread hypothesis. Our study demonstrated the potential of population genomic approaches via international partnership to address global emerging pest threats and biosecurity challenges.

[1] CSIRO Black Mountain Laboratories, Clunies Ross Street, Canberra ACT 2602, Australia. [2] CSIRO Australian Centre for Disease Preparedness, Geelong, VIC, Australia. [3] CSIRO FD McMaster Laboratories, New England Highway, Armidale NSW2350, Australia. [4] DGIMI, Université Montpellier, INRAE, Montpellier, France. [5] College of Plant Protection Nanjing Agricultural University, Nanjing, China. [6] Gordon Institute, University of Cambridge, Cambridge CB2 1QN, UK. [7] Universidade Federal de Goiáss, Escola de Agronomia, Goiânia, GO, Brazil. [8] National Crops Resources Research Institute, Namulonge, Kampala, Uganda. [✉]email: weetek.tay@csiro.au

Long an important pest of agriculture in its native New World range, the fall armyworm (FAW) Spodoptera frugiperda was first reported in West Africa (Nigeria, São Tomé and Príncipe) in early 2016[1], followed by confirmation across central (Congo[2]; Togo[3], Southern[4] and Eastern[5]) sub-Saharan Africa between 2017/2018[6]; the Middle East (Yemen[7]), India[8,9] and surrounding nations, Myanmar[10] and Thailand[11] (August and December 2018), followed by Southern China (Yunnan Province) in early January 2019[12–14]. Detections of FAW since January 2019 have gathered speed: south-ward to Malaysia (March 2019) and Indonesia (Sumatra, April 2019; Java, July 2019; Kalimantan July 2019); Hong Kong (April 2019), Taiwan (May/June 2019); Laos and Vietnam (April 2019[15]), the Philippines (June 2019[16,17]), South Korea (June 2019), and Japan (June 2019)[18]. Within China, the FAW has been reported in a northward expansion pattern from Yunnan to 18 provinces by July 2019[19–21]. As of September 2021, over 70 African and Asian nations have reported FAW[22]. In January 2020, FAW was trapped in Australia's special biosecurity zone in the Torres Strait islands of Saibai and Erub, and confirmed on 3 February 2020, and on mainland Australia in Bamaga on 18 February 2020[23,24].

This chronologically ordered eastward spread of detections led to a widely adopted assumption[25] that the FAW was actually spreading west-to-east across and then from Africa. Based on the detection timeline, predictive simulations that assumed human-assisted spread, in particular, eggs on aircraft carrier surfaces[26] and from agricultural trade (e.g., associated with various fresh agricultural commodities including asparagus, capsicum, Solanum melongena, S. macrocaropon; with cut flowers, and fresh cuttings[27]) associated with egg and larval stages, have modelled this very vagile pest's movement from the east coast of America/ the Greater Antilles to West Africa[3] between Central and Southern America and Africa, and between Africa and Asia[28]). Movements of soil from countries known to have FAW into the EU are prohibited and were put in place to limit accidental introductions of pupae[27]. The human-assisted spread model[28] was also used to warn China and South East Asian nations of imminent impact by FAW following confirmation of the pest in India[29]. This model further forms the basis of international research efforts to track the movement, including using molecular tools to examine invasion biology (e.g.,[3,30,31]), and simulations to model long-distance dispersal (e.g.,[28,32,33]). Indeed a meteorological data-based simulation study concluded the Yunnan FAW populations originated from Myanmar, consistent with FAW being officially reported earlier in Myanmar[34,35] than in China[19]. Other work has examined the impact and implications for global plant health and agricultural biosecurity (e.g.,[12,36]), integrated pest management (IPM) and bioeconomics[37–39], and insecticide resistance[21,31,40].

Genetic studies on the spread of FAW have focussed on single genes on the mitochondrial genome, occasionally supported by a single partial nuclear gene marker. These markers have been widely used because, throughout much of the native range, FAW populations consist of two morphologically identical host races, the rice-preferred and corn-preferred S. frugiperda ('Sfr' and 'Sfc', respectively), that have also been considered as potential sister species[5,41,42] that exhibited low but significant genomic variation (~1.9%)[30]. These two host races are supported by phylogenetic analyses based on nuclear and mitochondrial DNA genomes[30], and partial mitochondrial DNA genes (e.g.,[1,5,9,13,41]). The distribution of these Sfr and Sfc populations in their New World range has only recently been investigated based on partial mitochondrial and nuclear genes[43], while at the whole genome level they are less well-understood. Genotypes from both host races/ sister species are present in the invasive populations (e.g.,[3,44–46]). Since 2010[47,48] and especially in recent times during the FAW

range expansion[13,31,45,49], the partial Triose Phosphate Isomerase (Tpi) gene on the Z-chromosome has been adopted to clarify the Sfc/Sfr host race status. The Tpi marker relies on the presence of a single nucleotide polymorphic (SNP) marker at position 183[48,49] to differentiate between corn- or rice-preferred FAW. Similarly, inconclusive host preferences based on the mtCOI gene marker also detected Sfc and Sfr on corn-host plants (e.g.,[5]). Contrary to the introduction patterns of the noctuid H. armigera in South America[50] which showed high genetic diversity[51–53] similar to that reported for global H. armigera populations[54–56], the current global partial mtCOI signatures of both Sfc and Sfr have each been consistent with a single 'bridgehead effect'[57] introduction, which, when considered together with the Tpi locus, was suggested to likely have a Florida/Greater Antilles source population[3].

What is missing from current research into the spread of FAW is an analysis of broader genomic evidence. Genome-wide SNP markers aligned to well-annotated genomes can provide powerful genomic evidence for understanding introduction pathways[58] and eliminate candidate populations[59] as well as to elucidate hybrid signatures[60]. Furthermore, under the current assumption (i.e., a west-to-east spread involving a single western African invasive bridgehead population), subsequent invasive populations would share the founding population's genomic signatures, since as the more evolutionarily parsimonious scenario, only a single genetic change to match between the environment and the bridgehead invasive individuals would be needed[57].

In this study, we provide an assessment of global FAW movement history based on genomic data that incorporates populations from Northern, Central, and Southern Americas, and the Caribbean (i.e., representing the original population range), Western and Eastern Africa, and Western and Eastern Asia, representing the pest's Old World expansion. Here we reveal a multi-locus invasion that is likely independent of the reported detection patterns and their timelines, and provide genomic-based evidence to support multiple introductions of the FAW into the Old World, with movements of FAW detected between Asia and Africa. We also re-evaluated the pest's global spread directionality to highlight implications in the future management of FAW, and the need for ongoing global agricultural biosecurity research and cooperation to improve preparedness for emerging invasive agricultural pest issues.

## Results

Across the native and invasive ranges, FAW individuals have been classified into rice- or corn-preferred strains, either based on the partial mtCOI gene, or through the TPI partial gene from the z-chromosome. Due to the non-recombinant and maternally inherited mode of the mitochondrial DNA genome (cf. biparental and recombinant nuclear genome), it is also possible to infer the minimum number of unique female founders responsible for establishing the invasive populations, as well as their likely native population origin/s through matching between mitochondrial genomes. We first examined the mitochondrial genome diversity in the invasive range to determine minimum maternal lineages, and an overview of the strain composition in both native and invasive populations. Of the 197 FAW individuals sequenced (Supplementary Data 1), 102 were from the native New World range and 95 from the invasive Old World range (Fig. 1). From the pest's native range, we detected 25 'rice' mitochondrial genome (i.e., mitogenome) haplotypes, and 51 'corn' mitogenome haplotypes. All FAW from Mexico and Peru had the 'corn' mitogenome while FAW from Guadeloupe and French Guiana were all 'rice' mitogenomes. Of the FAW from the invasive range nine 'corn' and 'rice' mitogenome haplotypes were identified; one

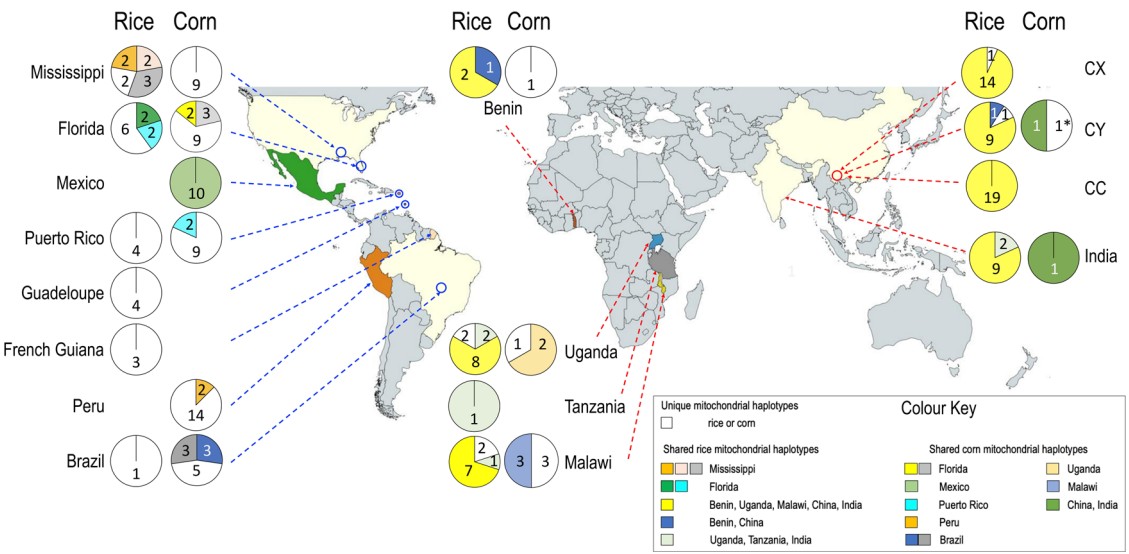

**Fig. 1 Native New World and invasive Old World FAW populations, and proportions of rice-strain and corn-strain mitochondrial DNA haplotypes from 15,059 bp of the mitochondrial DNA genomes.** New and Old Worlds' FAW populations and proportions of mitochondrial DNA haplotypes based on 15,059 bp of the mitochondrial DNA genomes and excluding four regions of low complexity. For the New World 'rice' FAW, a total of 20 unique mitogenome haplotypes (represented by the white colour proportion of each pie chart), and 11 non-unique mitogenome haplotypes were detected (i.e., a total of 25 mitochondrial haplotypes in rice FAW in the New World). For the 'corn' mitogenomes, 46 unique haplotypes were detected from the native range, while 25 corn FAW individuals shared a total of seven haplotypes (i.e., a total of 46 + 7 = 53 mitochondrial haplotypes). In the invasive range, six unique 'rice' mitogenomes (i.e., white portion of the pie charts, representing two individuals from Uganda, two individuals from Malawi, and two individuals from China (CY, n = 1; CX, n = 1) and three shared mitogenomes (i.e., dark blue, yellow, pale green) were detected from 76 individuals from Africa (n = 22), India (n = 11) and China (n = 43). For the 'corn' FAW from the invasive range, six unique mitogenome haplotypes (i.e., white portions of pie charts) and three non-unique mitogenome haplotypes (pale orange, pale blue and dark green) were detected, although only one individual each from China and India shared a common mitogenome (represented by dark green). With the exception of white colour representing unique mitogenomes, colour schemes are otherwise independent between 'corn' and 'rice' mitogenome haplotypes. China FAW populations from Yunnan Province of Cangyuan (CC), Yuanjiang (CY), and Xinping (CX) are indicated. One pre-border FAW intercepted in December 2016 from cut flowers that originated from Yunnan China (CH06) with a unique corn mitogenome is indicated with '*' (placed together with the CY corn pie-chart). Numbers within pie-charts indicate individuals for each mitogenome haplotype.

of the 'corn' mitogenome haplotypes (represented by green colour, Fig. 1) was shared between CY and Indian individuals. No African corn mitogenome haplotypes were shared with Asian FAW populations. In contrast, 83% (i.e., 68/82) of African and Asian FAW with 'rice' mitogenomes shared a common haplotype (represented by the yellow colour, Fig. 1). FAW individuals from China and Benin shared a rare rice mitogenome haplotype (blue colour haplotype), and individuals from Uganda, Tanzania, Malawi and India shared one (i.e., light green colour) haplotype (Fig. 1). In general, the high diversity of haplotypes in both 'rice' and 'corn' in the native range and the lack of diversity in the invasive range is consistent with patterns observed in invasive populations that have a relatively small number of founders.

**Mitochondrial DNA genome phylogeny.** The trimmed (15,059 bp) mitochondrial DNA genomes of all individuals identified two distinct clades that corresponded to the 'rice-preferred' and 'corn-preferred' clusters (Fig. 2). Based on the near-complete mitogenome phylogeny, a minimum of four and five introduction events were associated with the 'rice' and 'corn' maternal lineages, respectively (Fig. 2). Except for the 'corn' specimen (CH06) from Yunnan that clustered strongly with an individual from Mississippi (UM04) within a clade (node support:80%) that included also North and South Americas individuals, all 'corn' individuals from the invasive range (i.e., MW26, BE30, MW01, MW06. IN12, MW16, UG03, UG06) clustered weakly with individuals from Florida. Similarly, apart from the Benin individual (i.e., BE01), all remaining 'rice' FAW from the invasive range also clustered weakly with individuals from

Florida. Therefore, the likely origins of the Old World invasive 'corn' and 'rice' FAW remained inconclusive, while divergent mitochondrial genomes nevertheless supported multiple introductions to underpin the current invasive Old World FAW populations.

**Nuclear SNP phylogeny.** The ML phylogeny based on 870 unlinked and neutral SNPs revealed four distinct clades (clades I, II, III, IV; Fig. 3) across the sampled populations. Native and invasive individuals were a component of each clade which enabled a side-by-side comparison of population structure. Members within each clade were grouped with high (90–96%) bootstrap branch node support values. Clade I included the majority of the invasive FAW individuals from China (CX, CY, CC), India (IN), Uganda (UG), and Benin (BE) as well as individuals from Brazil. Overall, subclades within Clade I indicated unique genomic signatures between the CC and CY/CX populations. Indian and African populations (i.e., Uganda, Benin) were scattered among the CC and CY/CX populations. This interspersed clustering of subclades from Chinese, African and Indian populations suggest a complex FAW spread across the Old World, with some of the China CY individuals potentially sharing a New World origin similar to the Brazil rCC (i.e., 'BR' code, Fig. 3 Clade I) individuals.

Clade II, which is phylogenetically most closely related to Clade I, is dominated by individuals from Mississippi. Within this clade, individuals from China (i.e., CX), Uganda, Benin and India are also present, indicative of likely separate introductions of FAW from the population(s) with genetic similarity to the Mississippi

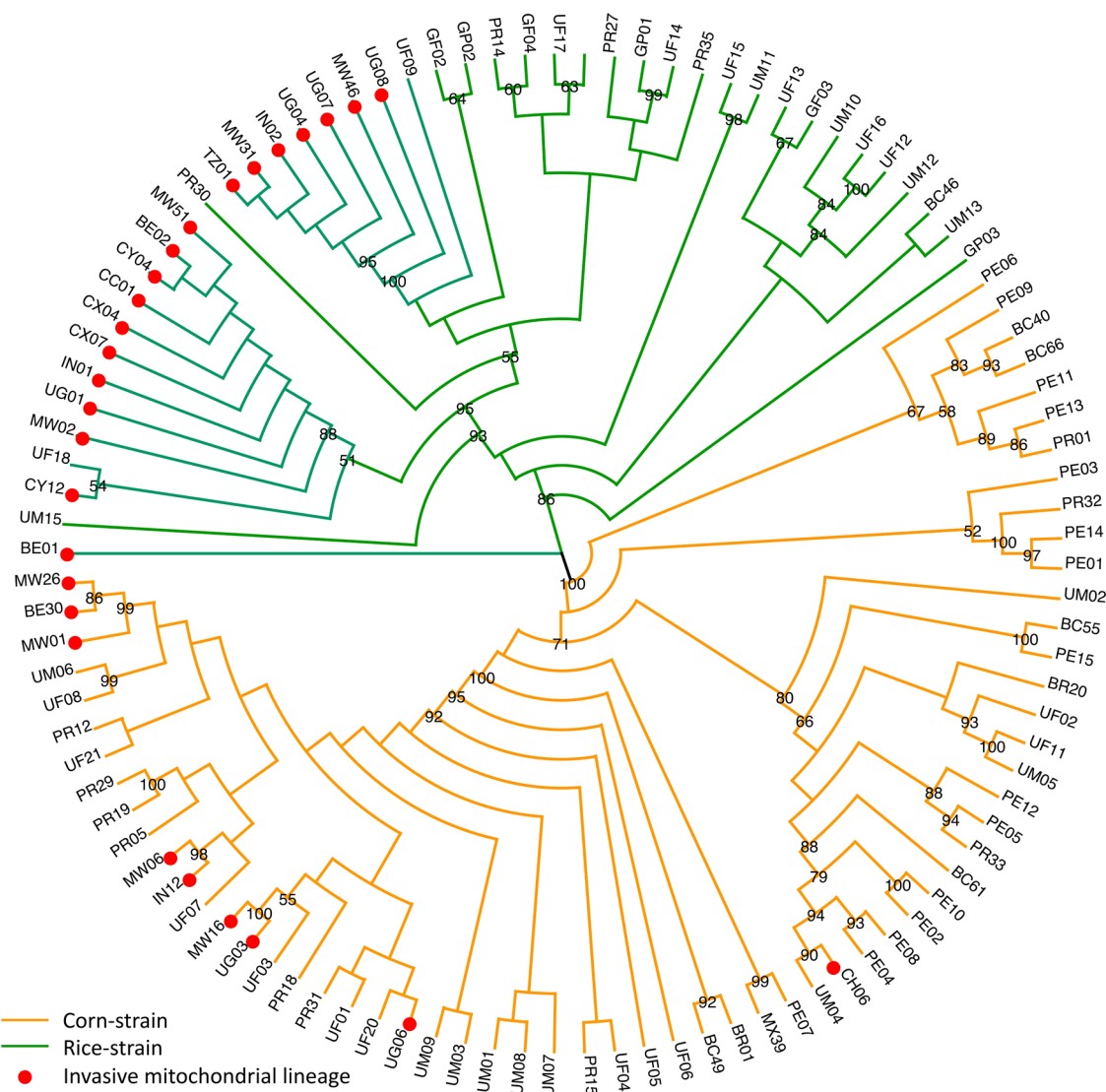

**Fig. 2 Partial (15,059 bp) mitochondrial genome Maximum Likelihood phylogeny, showing a clear dichotomy between the rice-strain (green branches) and corn-strain (orange branches)** *Spodoptera frugiperda.* **Invasive haplotypes from multiple maternal lineages are indicated by red dots.** FAW maximum likelihood phylogeny was constructed using IQ-Tree based on 15,059 bp partial mitochondrial genome with edge-linked partition for the 13 protein-coding genes and excluding four regions of low complexity. Node support is estimated from 1000 bootstrap replications, node support values are shown for ≥50%. 'Rice' clade is indicated by branches in green (native range) and 'Corn' clade is indicated by branches in orange (native). Unique haplotypes from all populations are included. Country codes are UF (USA-Florida), UM (USA-Mississippi), PR (Puerto Rico), GP (Guadeloupe), GF (French Guiana), PE (Peru), MX (Mexico), BC (Brazil-CC), BR (Brzil-rCC), BE (Benin), UG (Uganda), TZ (Tanzania), MW (Malawi), IN (India), and four populations from China Yunnan Province (Australia pre-border interception (CH06); Cangyuan (CC), Yuanjing (CY), and Xinping (CX)). Invasive mitochondrial lineages are indicated by red dots.

population into the Old World. Clade III is represented by a separate Brazilian (i.e., 'BC') FAW population and the Peru FAW individuals. Invasive populations clustered within clade III were the Malawi FAW population, a single Tanzania and three Ugandan individuals, suggesting that these African FAW shared a similar origin that is different from other African (e.g., Benin, rest of Uganda) and Asian populations. The Ugandan population, in particular, appears genetically highly heterogeneous, indicating it too has mixed introduction backgrounds.

Clade IV is dominated by the Florida population and other Caribbean islands/Greater Antilles (e.g., Puerto Rico)/Lesser Antilles (e.g., Guadeloupe)/ Central American (e.g., Mexico), and parts of the northern region of South America (e.g., French Guiana) FAW populations. Clade IV contained a single invasive Chinese FAW (i.e., CH06). Taken as a whole, the nuclear SNP

phylogeny provides clear evidence for multiple introductions of FAW to the Old World, while identifying populations associated with the Mississippi and the Brazilian 'BR' populations as likely sources of invasive populations into the Old World. The source population for Malawi's FAW was likely population(s) from South America, currently represented by Peru/Brazil (BC) populations. Based on interception data, with the exception of a single unique FAW, Florida and the Greater Antilles do not appear to be likely sources for the current invasive populations in the Old World.

Our nuclear SNP phylogeny therefore clearly showed that the native range FAW populations could be classified based on their geographic origins. The unexpected direct phylogenetic relationship between the US Mississippi and Brazil 'BR' population, suggested potential movements of populations within North

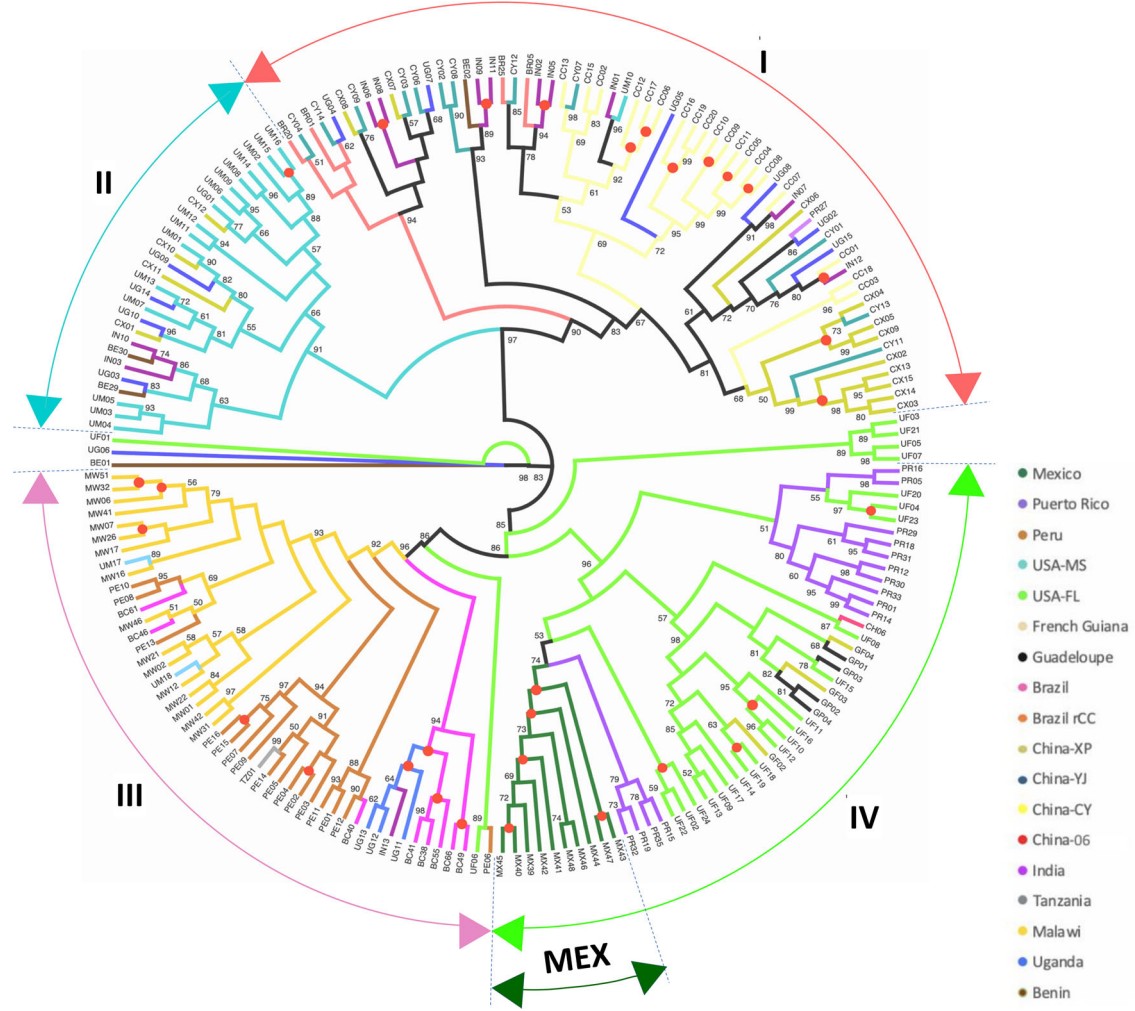

**Fig. 3 Maximum likelihood phylogeny of *Spodoptera frugiperda* populations from the native range of Northern America, Caribbean, South America, and the Old World invasive populations from Africa and Asia as inferred from 870 genome-wide SNP loci.** IQ-Tree with 1000 bootstraps replications to estimate node support for *Spodoptera frugiperda* populations from Northern America (Mississippi, Florida), Caribbean (Puerto Rico, Guadeloupe, French Guiana), and South America (Peru, Brazil), as well as *S. frugiperda* populations representing the Old World invasive range from Western Africa (Benin), Eastern Africa (Uganda, Tanzania, Malawi), and Asia (India, China). A total of 870 independent SNPs (i.e., unlinked) from non-coding regions distributed across the genome with no missing data were used. Populations are represented by unique colour schemes as indicated. Three populations of *S. frugiperda* from China Yunnan Province are Cangyuan (CC), Yuanjiang (CY), and Xinping (CX), and two populations of *S. frugiperda* from Brazil are Brazil-CC (BC) and Brazil-rCC (BR). Branch nodes with 100% bootstrap support are indicated by red dots. Bootstrap values of <50% are not shown. The legend shows branch colours of sampling countries.

America (i.e., Mississippi is not the wintering ground for FAW and represents the melting-pot for summer migrants from Texas and Florida[61] and between North and South America. Finally, an important overall finding was that our panel of neutral SNPs selected from whole-genome sequencing did not separate individuals based on 'corn' or 'rice' mitochondrial DNA genome signatures, nor did they support the host strain characterisation based on the *Tpi* partial gene marker.

**Genetic diversity.** Basic population diversity statistics for each population are listed in Table 1. Nucleotide diversity ($\pi$) varied across a narrow range (0.287–0.329), for the 870 variable and independent SNPs analysed, that included no invariant loci. No significant overall difference was observed between the native and invasive range populations. All populations showed higher average observed heterozygosity ($H_{obs}$) than the average expected heterozygosities ($H_{exp}$), both in the native and invasive ranges, with the highest $H_{obs}$ seen in the Malawi population. Negative $F_{IS}$ values for all populations were consistent with $H_{obs}$ being higher

than $H_{exp}$, and suggested systematic avoidance of consanguineous mating[62] within FAW subpopulations as a whole. The lower expected heterozygosity in all these populations (i.e., $H_{obs} > H_{exp}$; see[63]) is most likely indicative of the recent mixing of previously distinct populations and does not support that these invasive populations originated from a single introduction (e.g.,[2,49,64,65]) or had undergone a recent bottleneck from the widely suggested recent western Africa arrival. It is more likely that they represent the result of multiple introductions into the invasive range, as already suggested by mitochondrial and nuclear SNP phylogenies (Figs. 2, 3) and PCA (Fig. 4). The observed heterozygosity excess detected for the native range populations may similarly be due to factors such as structure between these populations and the breaking of isolation through periodic migration among native populations. Consistent with these observations, a number of the populations including most from the invasive range also contained significant numbers of loci not in Hardy-Weinberg equilibrium (HWE). This was especially the case for the two largest Chinese populations (i.e., CY, CX), Malawi and Uganda, as well

**Table 1 Population statistics for native and invasive range *Spdoptera frugiperda* populations.**

| Pop. code | Pop. | No. samples | Avg. $H_{exp}$ | Avg. $H_{obs}$ | HWE, $P > 0.001$ | $F_{IS}$ | Nt diversity ($\pi$) |
|---|---|---|---|---|---|---|---|
| BC | Brazil-CC | 8 | 0.289 | 0.420 | 870 | −0.241 | 0.309 |
| BE | Benin | 4 | 0.274 | 0.408 | 870 | −0.179 | 0.313 |
| BR | Brazil-rCC | 4 | 0.263 | 0.396 | 870 | −0.178 | 0.301 |
| CC | China-CY | 19 | 0.282 | 0.400 | 796 | −0.262 | 0.289 |
| CH | China-H06 | 1 | | | | | |
| CX | China-XP | 15 | 0.293 | 0.416 | 837 | −0.263 | 0.303 |
| CY | China-YJ | 12 | 0.284 | 0.405 | 870 | −0.248 | 0.296 |
| GF | French Guiana | 3 | 0.247 | 0.375 | 870 | −0.138 | 0.296 |
| GP | Guadeloupe | 4 | 0.245 | 0.359 | 870 | −0.152 | 0.279 |
| IN | India | 12 | 0.289 | 0.403 | 870 | −0.239 | 0.301 |
| MW | Malawi | 16 | 0.319 | 0.461 | 838 | −0.303 | 0.329 |
| MX | Mexico | 10 | 0.265 | 0.403 | 870 | −0.263 | 0.279 |
| PE | Peru | 16 | 0.319 | 0.456 | 848 | −0.295 | 0.329 |
| PR | Puerto Rico | 15 | 0.288 | 0.404 | 845 | −0.251 | 0.298 |
| TZ | Tanzania | 1 | | | | | |
| UF | USA-FL | 24 | 0.281 | 0.383 | 810 | −0.242 | 0.287 |
| UG | Uganda | 15 | 0.305 | 0.428 | 843 | −0.266 | 0.315 |
| UM | USA-MS | 18 | 0.320 | 0.453 | 820 | −0.293 | 0.329 |

The native range FAW populations are: USA-Florida (UF), USA-Mississippi (UM), Brazil-rCC (BR), Brazil-CC (BC), Puerto Rico (PR), Guadeloupe (GP), French Guiana (FG), Peru (PE), Mexico (MX) and the invasive range FAW populations are Benin (BE), Uganda (UG), Tanzania (TZ), Malawi (MW), India (IN), and China (CH, CC, CY, CX). See Supplementary Data 1 for sample and population details, and see "Methods" for details of how the statistics were calculated. Neutrality tests (Tajima's D; Fu & Li's D*; Supplementary Data 2) were only calculated for populations with at least four samples. Nucleotide diversity (Nt diversity, $\pi$) was calculated using Stacks only for the variant loci analysed and no window size specified. Avg. $H_{exp}$: average expected heterozygosity, Avg. $H_{obs}$: average observed heterozygosity; $F_{IS}$: inbreeding coefficient. Our high nucleotide diversity ($\pi$) estimates reflected the result of estimating based on limited (i.e., 870) polymorphic SNPs across the genome and will have comparative value for future studies that utilised similar sets of SNPs.

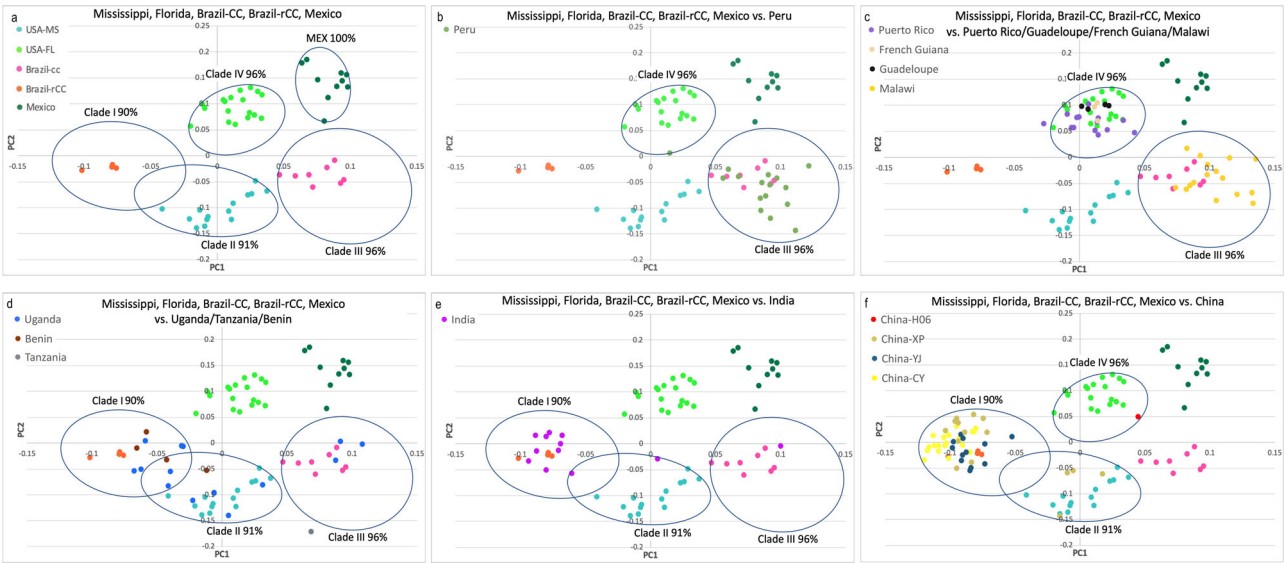

**Fig. 4 Principal Component Analyses of native and invasive *Spodoptera frugiperda* populations based on 870 neutral SNP loci.** Principal Component Analyses of native and invasive FAW populations based on 870 neutral and unlinked SNP loci. **a** The five clusters of native FAW populations (identified also from the genome-wide SNP phylogeny in Fig. 3). Circles indicate confidence as shown in Fig. 3. **b** Peru individuals clustered overall with Brazil-CC population (Clade III; pink colour) but also overlapped Florida population (Clade IV, light green colour). **c** Puerto Rico (purple colour), Guadeloupe (black colour) and French Guiana (wheat) overall clustered with Florida population with 96% confidence, while the invasive FAW population from Malawi (yellow colour) clustered in Clade III with Brazil-CC and Peru with 96% confidence. **d** PCA of Uganda population (blue colour) indicated the population was scattered across Clades I, II and III, Benin individuals (Saddlebrown colour) fell within clades I and II, while Tanzania (Azure 4 colour) fell just outside of 96% confidence of Clade III. **e** Indian FAW individuals showed similar clustering patterns as the Ugandan individuals, being found in Clades I, II, and III. **f** Chinese FAW populations were predominantly clustered within Clade I, with few CX individuals also found within Clade II. No individual from China was found in Clade III, while one individual originating from Australia's pre-border inspection program was clustered with the Florida population (Clade IV) at 96% confidence. No invasive populations were clustered with the Mexican population. Colour codes for populations as provided in Fig. 3.

as for several native range populations; many of the populations studied, therefore, appear to result from the recent mixing of previously separated populations. Approximately half the number of loci departed significantly from HWE in the global population (i.e., 437 of the total 870) and highlighted the complex population structure in both native and invasive ranges. For example, there

have been limited studies of seasonal migratory behaviours of native FAW populations between South, Central, and North Americas, with populations from South America, often found to have a high inbreeding coefficient (e.g.,[66]). Migratory patterns in hybrid populations across Africa, Asia, Southeast Asia, and Oceania, have remained largely unknown, and unlikely to

represent panmictic populations due to multiple origins of founding populations, at least for the African, Indian, and Chinese FAW populations analysed.

**Population structure and migration.** Multivariate Principal Component Analysis (PCA) of the 197 individuals in the native and invasive populations based on the 870 neutral and unlinked SNP loci showed the individuals to largely cluster according to their populations, as observed in the phylogenetic analyses (above). The native FAW populations formed five clusters (Fig. 4a), while native range samples showed FAW from Peru to overall cluster with the Brazil-CC population (code 'BC') but also overlapping with those from Florida (Fig. 4b). Samples from Puerto Rico, Guadeloupe and French Guiana tended to cluster with the Floridian population with 96% confidence (Fig. 4c). This panel also showed the invasive FAW population from Malawi clustering with Brazil-CC and Peru in Clade III, with 96% confidence. The Ugandan population was scattered across Clades I, II and III (Fig. 4d) while the Benin individuals fell within clades I and II and that from Tanzania fell just outside of 96% confidence of Clade III. Indian FAW individuals showed similar clustering patterns to those of Ugandan individuals, being found in Clades I, II, and III (Fig. 4e). The Chinese FAW populations were predominantly clustered within Clade I, with a few XP individuals, also found within Clade II (Fig. 4f). No individual from China was found in Clade III, while one individual (CH06) was clustered with the Florida population (Clade IV) at 96% confidence. We did not identify any invasive population to cluster with the Mexican population.

Pairwise genetic differentiation estimates ($F_{ST}$) between populations varied significantly (Table 2). The Mexico and Brazil-rCC (BR) populations showed strong genetic differentiation with all other populations, while the Brazil population showed low genetic differentiation that could suggest gene flow with both Peru and US Mississippi (UM) populations. There was a lack of population substructure, especially between invasive range populations which suggests varying levels of gene flow. Significant population substructure was detected between Peru and invasive FAW populations from China-CY, China-XP and China-YJ, and India, while $F_{ST}$ estimates indicated low genetic differentiation between African populations (Benin, Tanzania, Uganda, and Malawi), thereby suggesting some level of movements within African populations.

**Admixture analysis.** Analysis of populations using Admixture showed structure evident at K values from 3 to 5 (Fig. 5). At $K = 3$, a total of six Chinese individuals from the CY and YJ populations appeared to be non-admixed (red dots). Similarly, at $K = 4$, three of these six FAW individuals remained non-admixed as also indicated (red dots). However, at $K = 5$, the number of non-admixed individuals nearly doubled compared with $K = 3$. No other FAW individuals from the invasive range otherwise showed non-admixed genomic signatures irrespective of the K-values of 3, 4 or 5. The Malawi FAW individuals share very similar admixture patterns as FAW individuals from Peru and Brazil-CC (i.e., 'BC') populations. This shared admixed profile between Malawi and Peru/BC populations is especially clear at $K = 5$, which also enable clearer visualisation of the Tanzanian individual and selected Ugandan individuals (e.g., UG11, UG12, UG13) as also having similar admixture profiles as Malawi individuals (see also Figs. 3 and 4c, d).

Admixture analysis of native populations of FAW showed that the majority of individuals have admixed genomic signatures. The exceptions are individuals from Florida (e.g., UF19, UF09, UF12, UF16), and Guadeloupe (GP02, GP04) at predominantly $K = 4$

**Table 2 Population genetic differentiation via pairwise $F_{ST}$ estimates between native and invasive range *Spodoptera frugiperda* populations.**

| | BE | BC | BR | CC | CH | CX | CY | GF | GP | IN | MW | MX | PE | PR | TZ | UF | UM | UG |
|---|---|---|---|---|---|---|---|---|---|---|---|---|---|---|---|---|---|---|
| BE | N/A | *** | *** | *** | | *** | *** | | | *** | | *** | | | | *** | | *** |
| BC | 0.027 | N/A | *** | *** | | *** | *** | | | *** | ^ | *** | | +‡ | | *** | *** | *** |
| BR | 0.071 | 0.039 | N/A | *** | *** | *** | *** | *** | *** | *** | *** | *** | *** | *** | *** | *** | *** | *** |
| CC | 0.012 | 0.05 | 0.09 | N/A | | *** | *** | *** | ^ | | ‡ | *** | | | | | | |
| CH | 0.029 | 0.02 | 0.124 | 0.057 | N/A | | | | | | | *** | | | | | | |
| CX | 0.019 | 0.05 | 0.08 | 0.02 | 0.049 | N/A | | | | | | *** | | | | *** | *** | |
| CY | 0.006 | 0.04 | 0.08 | 0.008 | 0.052 | 0.012 | N/A | | ^ | | | *** | ^ | | | *** | *** | |
| GF | 0.036 | 0.039 | 0.088 | 0.048 | 0.035 | 0.053 | 0.002 | N/A | ^ | | +^ | *** | ^ | | | | *** | |
| GP | 0.041 | 0.048 | 0.086 | 0.06 | 0.032 | 0.06 | 0.05 | 0.037 | N/A | | +^ | *** | | | | *** | *** | |
| IN | 0.004 | 0.04 | 0.07 | 0.008 | 0.024 | 0.012 | 0.006 | 0.039 | 0.043 | N/A | | *** | *** | | | *** | *** | |
| MW | 0.007 | 0.02 | 0.05 | 0.02 | 0.088 | 0.03 | 0.02 | 0.09 | 0.05 | 0.02 | N/A | *** | *** | ^ | | *** | *** | |
| MX | 0.07 | 0.07 | 0.1 | 0.08 | 0.018 | 0.05 | 0.08 | 0.037 | 0.09 | 0.07 | 0.07 | N/A | *** | *** | | *** | *** | *** |
| PE | 0.02 | 0.011 | 0.042 | 0.05 | 0.024 | 0.05 | 0.019 | 0.037 | 0.04 | 0.04 | 0.014 | 0.06 | N/A | | | *** | *** | *** |
| PR | 0.018 | 0.051 | 0.06 | 0.03 | 0.024 | 0.076 | 0.069 | 0.081 | 0.098 | 0.02 | 0.028 | 0.05 | 0.03 | N/A | | *** | *** | *** |
| TZ | 0.041 | 0.03 | 0.14 | 0.071 | N/A | 0.03 | 0.03 | 0.008 | 0.014 | 0.047 | 0.007 | 0.132 | 0.019 | 0.066 | N/A | | | |
| UF | 0.016 | 0.018 | 0.06 | 0.03 | 0.016 | 0.05 | 0.05 | 0.032 | 0.05 | 0.02 | 0.03 | 0.06 | 0.03 | 0.01 | 0.062 | N/A | *** | *** |
| UM | 0.034 | 0.03 | 0.042 | 0.05 | 0.026 | 0.021 | 0.013 | 0.041 | 0.05 | 0.04 | 0.02 | 0.06 | 0.01 | 0.03 | 0.021 | 0.03 | N/A | *** |
| UG | 0.003 | 0.03 | 0.06 | 0.02 | 0.025 | 0.021 | 0.013 | 0.032 | 0.043 | 0.011 | 0.001 | 0.07 | 0.02 | 0.02 | 0.011 | 0.03 | 0.03 | N/A |

Population codes are: Benin (BE), Brazil-CC (BC), Brazil-rCC (BR), China Cangyuan (CC), China-06 (CH), China Xinping (CX), China Yuanjiang (CY), French Guiana (GF), Guadeloupe (GP), India (IN), Malawi (MW), Mexico (MX), Peru (PE), Puerto Rico (PR), Tanzania (TZ), USA Florida (UF), USA Mississippi (UM), Uganda (UG). The $F_{ST}$ values are given in the lower left half of the table, and the p-values are given in the upper right half of the table (***$p \ll 0.001$; ++$p \leq 0.01$; ^$p \leq 0.05$). Both Tanzania and China-H06 populations consisted of one individual each and their pairwise $F_{ST}$ was therefore not estimated.

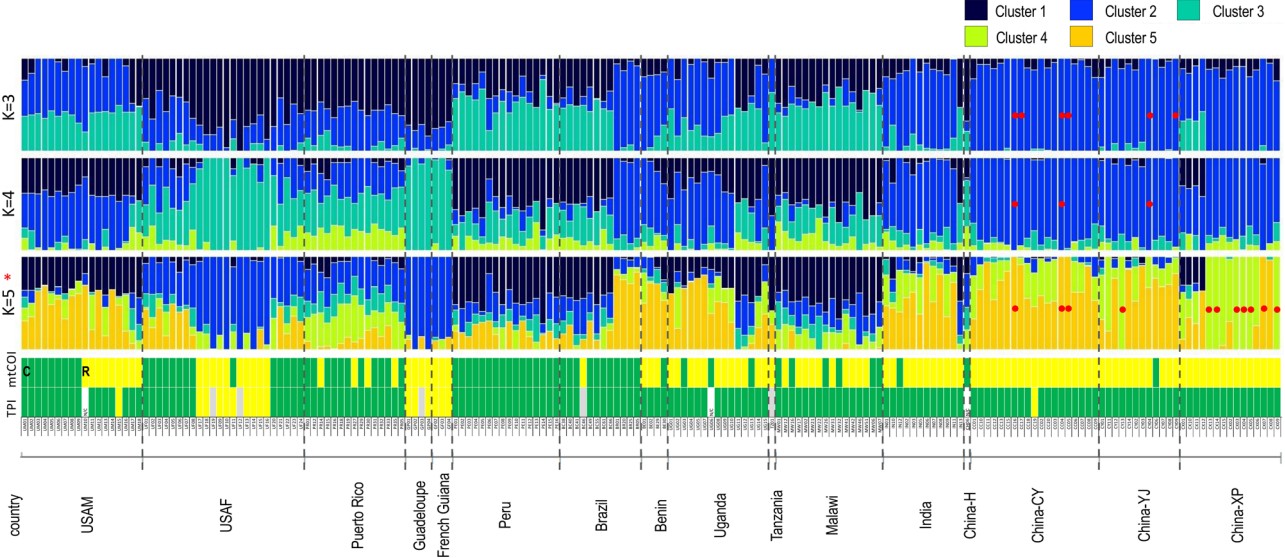

**Fig. 5 Admixture analysis of invasive and native *Spodoptera frugiperda* populations based on 870 neutral SNP loci, and individual host strain identity based on the *Tpi* and the mt*COI* markers.** Admixture analysis based on K = 3 (Cross-Validation Error (CVE): 0.375), K = 4 (CVE: 0.377), and K = 5 (CVE: 0.380)). Populations 'USAM' and 'USAF' are from Mississippi and Florida, respectively. Populations from China were from Cangyuan (CY), Yuanjiang (YJ), Xinping (XP) in Yunnan Province. Individuals in the invasive ranges that lacked the signature of admixture are indicated by red dots. Corn- ('C') or rice- ('R') preferred plant hosts are identified based on mtCOI as per Dumas et al.[41], and by *Tpi* SNP approach as per Nagoshi[48] are indicated by green or yellow bars, respectively. Specimen ID's and sampling countries are as labelled. 'N/C' for *Tpi* indicates no coverage. Grey bars indicate individuals with heterozygous *Tpi* SNPs.

and K = 5. Interestingly, these individuals with non-admixed genomic signatures (at either K = 3, 4 and/or 5) also possessed the rice mitogenome haplotypes (Fig. 5). This observation is similar to that observed for the non-admixed Chinese individuals that have mitogenomes which also exhibited the rice haplotypes. Admixture analysis also revealed most Mexican individuals as having non-admixed genome patterns and with the corn mitogenome haplotypes (Supplementary Fig. 1a, b). As with the SNP phylogeny above, a comparison of the admixture patterns to mitogenomes and the *Tpi* locus of native and invasive FAW populations failed to find evidence to support FAW host-strain characterisation. The genome admixture signatures of FAW across its African and Asian invasive range supported a complex pattern for multiple introductions. For example, given the highly admixed genomic patterns detected in African and Indian individuals, it is unlikely that matings between these admixed populations would lead to individuals with non-admixed genomic signatures in China, unless there was some very strong selection pressure acting across specific genomic regions of these selected CY, CC, and CX individuals.

**Admixture networks.** To explore the population substructure revealed by the admixture analysis in relation to the ML clusters obtained from phylogeny and PCA, we performed network analysis using the plotAdmixture function in the NetView R package. The ML network of individuals belonging to each of the specified populations is shown in Fig. 6. The four major clusters, I–IV, correspond to those shown in the ML tree (Fig. 3). Individuals from some populations were shown to be spread across multiple clades, e.g., PR, UF and UM from the native range and IN, BE and CX from the invasive populations. Of the populations in the invasive range, those from China were found predominantly in cluster I, with some CX individuals in cluster II and the single CH06 individual in cluster IV.

Plotting admixture proportions at K = 5 on this network showed the different populations from China that predominantly comprise Cluster I each have distinct admixture profiles that are

shared with those of individuals from Uganda and India. In cluster II, China-XP (CX), India (IN), Benin (BE), and Uganda (UG) formed networks with USA-Mississippi (UM) individuals. In Cluster III, all Malawi (MW) individuals and various Ugandan (UG) individuals and the single Tanzanian (TZ) individual formed a network cluster with Peru (PE), Brazil-CC (BC), and some USA Florida (UF) individuals. In cluster IV, only one Chinese FAW (CH) was found to group to this predominantly Caribbean/Central America FAW group (consisting of UF, Puerto Rico (PR), French Guiana (GF), Guadeloupe (GP), and Mexico (MX) FAW individuals).

**Directionality of gene flow analysis using divMigrate.** Analysis of the directionality of gene flow (i.e., relative directional migration) between populations using divMigrate enabled investigation of possible introduction pathways leading to the complex population substructure patterns seen in the above analyses. The most significant directional gene flow signatures seen were from all three Chinese populations (i.e., CX, CY, CC) into Malawi and from the Cangyuan (CC) population into Uganda (Fig. 7). Significant gene flow from Florida (UF) and from Puerto Rico (PR) into the Mississippi (UM) FAW population, which the above (e.g., Figs. 3, 4a, 5, 6; Supplementary Fig. 2) had shown to be distinct was also detected. No evidence was found for directional gene flow from any of the populations studied into China, nor any from or into India. Together with the Admixture results (Fig. 5), these results indicate the East African FAW populations likely originated from China, with some independent 'non-China' introductions also detected in Malawi. The Admixture signatures within the Ugandan FAW population suggested the presence of two genetically distinct FAW populations (Figs. 5, 6), one of which originated from Asia and involved genetic contribution from the Yunnan Cangyuan (CC) population (Fig. 7), as well as gene flow from Malawi (Fig. 5). While the Malawi population overall showed admixture patterns similar to Peru (Fig. 5) with the PCA showing the Malawi, Peru and Brazil-CC (BC) populations clustered together (Fig. 4b, c), directionality analysis

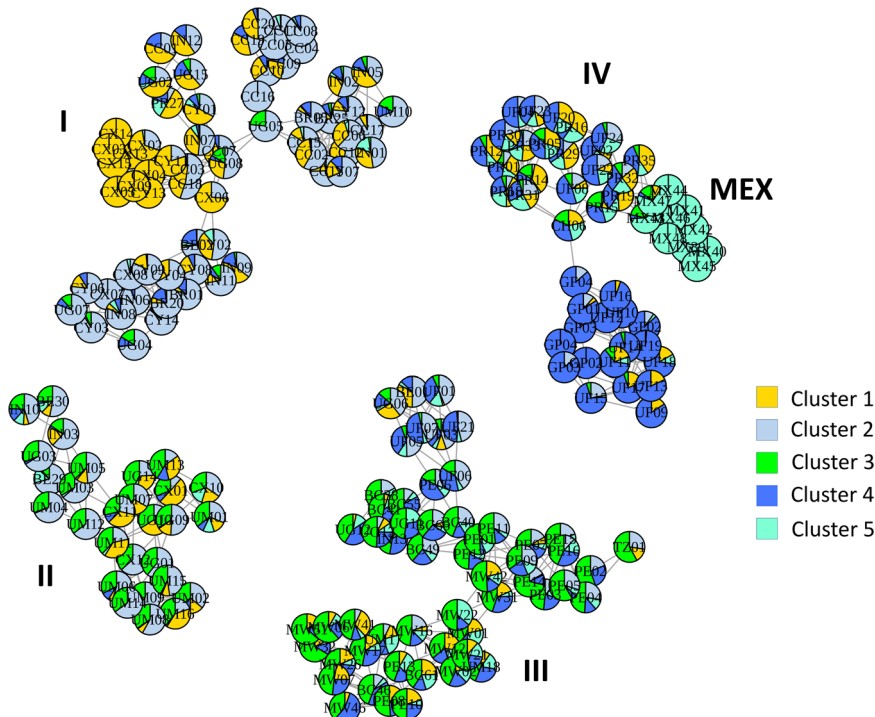

**Fig. 6 Maximum Likelihood distance network with admixture analysis of invasive and native *Spodoptera frugiperda* populations.** Maximum likelihood (ML) distance network with admixture analysis inferred from five genetic clusters (*K* = 5) presented as pie charts for each individual analysed. The network was drawn using the plotAdmixture function in the R package NetView[120, 121], and is based on a ML distance matrix calculated from the IQ-Tree shown in Fig. 3. using the R package ape[122]. The four major clusters, I–IV, correspond to those shown in the phylogenetic tree (Fig. 3). Individuals are identified by country codes as follows: China XP (CX), China YJ (CY), China CY (CC), China CH06 (CH), India (IN), Uganda (UG), Tanzania (TZ), Malawi (MW), Benin (BE), Brazil CC (BC), Brazil rCC (BR), Peru (PE), French Guiana (GF), Mexico (MX), Guadeloupe (GP), Puerto Rico (PR), USA-Florida (UF), and USA-Mississippi (UM). See Supplementary Data 1 for complete information about the individuals. Cluster I comprises predominantly different Chinese populations each with distinct admixture profiles but included also genetic cluster profiles of individuals from Uganda, India, Brazil-rCC (BR) and Puerto Rico. In cluster II, China-XP (CX), India, Benin, and Uganda formed networks with USA-Mississippi individuals. In Cluster III, all Malawi individuals and various Tanzania and Uganda individuals were grouped with Peru, Brazil-CC (BC), and selected USA-FL individuals. In cluster IV, only one Chinese FAW (CH) was found to a group to this predominantly Caribbean/Central America FAW group (consisting of USA-FL, Puerto Rico, French Guiana, Guadeloupe, and Mexico FAW individuals). Note that individuals sharing the same colour schemes do not necessarily have the same genetic content, and that the MEX group consisted only of individuals from Mexico showing little admixture with any other population.

indicated genetic contributions from all three Chinese FAW populations (Fig. 7).

## Discussion

The genomic analysis of FAW from native and invasive ranges contradicts recently published theories on the pathway, origin, and direction of the spread of this pest across the Old World. Neutral and unlinked genome-wide SNPs obtained from early stages of the FAW invasion showed, through population admixture analysis, ML distance network, and gene flow directionality analyses, that there were likely multiple introductions to both Africa and Asia. Studies to date have relied on analyses of limited partial mitochondrial DNA (e.g., partial *COI* and *CYTB*[5,46]); and the nuclear *Tpi* partial gene (e.g.,[45]) of various African, Asian and South East Asian invasive FAW populations, with comparisons to native New World FAW populations. These studies inferred the directionality of spread from the timing of official reporting to the FAO/IPPC, and described a single introduction of FAW to the Old World from an eastern American/Greater Antilles population, that spread rapidly across the sub-Saharan African nations, before moving to the Indian subcontinent via the Middle East, and then to South East Asia, and China[45].

Under the bridgehead effect of invasion scenario where subsequent successful invasive populations (e.g., FAW populations

from sub-Sharan Africa (south, central, east Africa); Asia (e.g., India, China)) originated via an intermediate and successful population (i.e., western African FAW population[1]), one should expect all populations to share the same genomic signatures as the bridgehead (i.e., western African FAW) populations[57]. Genome-wide SNP analyses in this present study, however, showed the populations in China and Africa to be genetically diverse and demonstrates strong evidence for a complex spread pattern across the Old World, including a substantial proportion of east-to-west movement, with populations from Asia as a potential source of invasive FAW populations in Africa (e.g., Malawi, Uganda), although our study lacked other populations (e.g., from Southeast Asia) that may be alternative sources of invasive FAW populations to Africa. The confirmation of FAW after reports by farmers of crop damage, i.e., in Nigeria and São Tomé and Príncipe in early 2016[1] and in northern and eastern Uganda since 2014[22], suggested that *S. frugiperda* was present in the African continent earlier, and given the genomic evidence reported here would suggest that the FAW was present in Asia and/or Southeast Asia prior to 2016.

Incidences of FAW attacking stem/leaf parts were reported from a farm producing turf grass for parks in Hanoi, Vietnam, between March and June (Spring/Summer seasons) of 2008, with heavy infestations reported between the months of April and May in 2008[67–69], as well as being reported as an agriculture insect

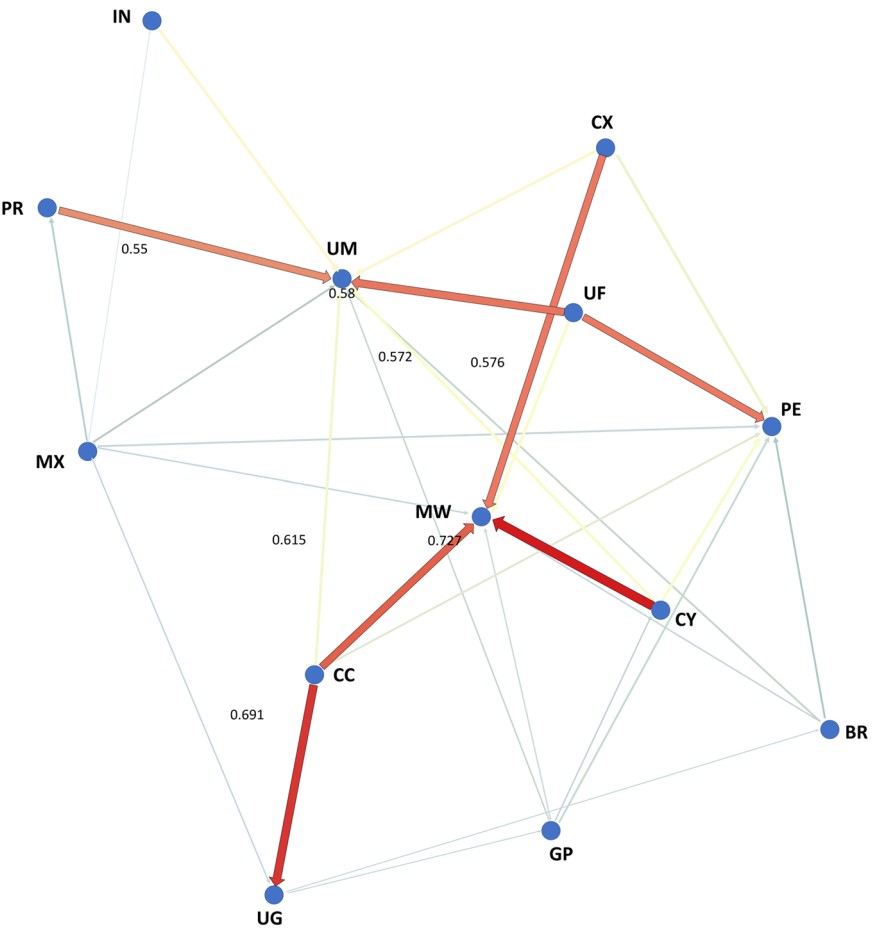

**Fig. 7 divMigrate Analysis inferred directionality of gene flow between native and invasive *Spodoptera frugiperda* populations.** Analysis using divMigrate to infer directionality of gene flow (i.e., relative directional migration) between New World native and Old World invasive *Spodoptera frugiperda* populations. The divMigrate analysis was run using the online server <https://popgen.shinyapps.io/divMigrate-online/>[122]. The analysis was performed with the $G_{ST}$ migration statistic of[127] and[128] at filter threshold = 3.0 and 1000 bootstrap replications to assess confidence with alpha value set at 0.05 (i.e., 95% confidence). Weighted values above 0.50 are indicated. Population codes are IN (India), PR (Puerto Rico), MX (Mexico), UG (Uganda), CC (China Cangyuan), CY (China Yuanjiang), CX (China Xinping), MW (Malawi), PE (Peru), GP (Guadeloupe), BR (Brazil-rCC), UM (USA Mississippi), and UF (USA Florida). High migration (i.e., gene flow; *sensu* Sundqvist et al.[124]) is seen from all three Chinese populations into Malawi and from Cangyuan (CC) to Uganda (UG). High migration from Florida and from Puerto Rico into the Mississippi FAW population is also detected.

pest in the production areas around Hanoi since 2008[70]. We also provided clear evidence for multiple introductions of this agricultural pest into Africa, demonstrating conclusively that the Malawian FAW population has a distinct genomic signature different from Chinese populations. The pre-border interceptions of FAW larvae (identified via morphological characters) that originated from countries outside of the Americas since prior to 2014[71] (although molecular diagnostics of these suspect FAW larvae will be required to provide definitive confirmation of such non-native range interceptions), the early detections and report of FAW in Asia/S.E. Asia (e.g., CH06[12] (GenBank MT897262); 2008 Hanoi outbreaks[69]), and the complex pattern of multiple introductions including potential North American origins for various Chinese FAW populations (e.g., this study for individual CH06; the Yunnan 'NJ05' Individual[72]), are consistent with the perceived rapid spread experienced across the African[73] and Asian continents[74].

Despite being one of the worst agricultural pests in the New World, there has been limited population genomic work on the FAW in their native ranges. Through our genome-wide SNP analyses, we have identified unexpected complexity in the FAW population structure in the New World. While the mitochondrial genome analysis confirmed the two canonical clades that have

long been suggested to define two strains with different host preferences, i.e., corn (*Sfc*) and rice (*Sfr*), the neutral nuclear SNP analyses showed a more complex population and genomic structure. FAW populations in the New World could be differentiated into at least five distinct groups that broadly followed the species' geographic distributions, and with no obvious pattern related to host race determination by mitochondrial or *Tpi* markers, providing the first genome-wide support for suggestions that these mitochondrial genomes (and the often associated *Tpi* marker) do not define any real population structure across the native range of FAW[43,75], while a lack of consistent correlation between host plant and mitochondrial genome in native range populations were observed[76]. Frequent hybridisation has been known to occur in the field (e.g.,[77]), and would also account for the observed pattern. Furthermore, African populations contained hybrids of F2 or even later generations, and mating time differences within the African populations were likely related to the differences in circadian gene expression previously identified in *Sfc* or *Sfr* populations in their native range[78]. Differences in mitochondria function could be directly related to host preferences[79], which could explain the absence of a correlation between the mitochondrial and nuclear genotypes, but this lack of genomic correlation that was also in part due to the non-

recombination nature for mitochondrial genome *cf.* nuclear genome, points clearly to the need of genome-wide studies in field populations, and that persistence to classify the invasive FAW populations into either 'rice' or 'corn' strains would contribute little in predicting crop damage, and may even hinder the management of these invasive FAW populations.

We detected directional migration from Florida and the Puerto Rican populations to the genetically distinct Mississippi one. This is consistent with findings based on mtCOI sequences that the Mississippi populations were established through seasonal migration from Texas and Florida[61]. There also seems to be evidence for a wider Caribbean population including Florida, Puerto Rico, Mexico, the Lesser Antilles (e.g., Guadeloupe) and the north-eastern region of South America (e.g., French Guiana). Mexican FAW formed a separate sub-clade within the Florida/ Greater Antilles/Lesser Antilles FAW group. Significant pairwise $F_{ST}$ estimates between Mexico and all native and invasive FAW populations and suggested population genetic differentiation that indirectly indicated limited gene flow. Northern Mexican populations have been shown to be similar to the Southern Texas overwintering population[61], and this is reflected by our finding that the Mexican population sits within the broader Caribbean clade that includes Florida. Across the native range, evidence of population substructure has been reported (e.g.,[66,80,81]), and a population genomic approach could help identify biological/ ecological factors that underpinned patterns of population differentiation, gene flow directionality, and prevalence of admixture to further assist with their management.

Our PCA on genome-wide SNPs identified the Brazilian FAW as two genetically distinct populations, with one population ('BC') being phylogenetically more closely related to the Peruvian FAW population, and the BR population which is phylogenetically more closely related to the Mississippi population. The Brazilian 'BR' population included individuals that had been found to have a novel 12 bp deletion mutation in the *ABCC2* gene[40]. The implications of the close phylogenetic relationship between the BR and Mississippi populations are great given that FAW is regarded as a major agricultural pest in Brazil[40], and the possible movements of alleles that could potentially underpin resistance, especially to Cry1F and Cry1A toxins, would add to the challenge of managing this pest in the Americas.

Genomic analyses in the present study support multiple introductions of FAW from different sources into Africa, rather than via a single western Africa introduction. Phylogenetic inference and PCA clearly identified the South American FAW population, as represented by the Peru/Brazil (BC) samples, as the likely source for the Malawi population, although this could also represent the 'bridgehead effect'[57] from other invasive FAW populations not yet included in the current analysis, such as other Asian/East Asian/South East Asian populations. Global movements of invasive pests, exemplified by the spread of FAW, and other agriculturally important pests (e.g., *H. armigera*[51,82]; the Harlequin ladybird *Harmonia axyridis*[83]; the whitefly *Bemisia tabaci* species complex[58]; the tomato leaf miner *Tuta abosulta*[84]) are timely reminders of the need for global coordination of enhanced biosecurity preparedness strategies that build on advancement in genomic research. The potential negative impacts of introductions of alien species include the introgression of genetic traits to local species through hybridisation[12,59,60,85,86]. Development of new trans-continental trade routes to increase economic growth between trading partners must therefore recognise these potential risks and take into consideration the biosecurity implications associated with the rapid spreading of highly invasive pests and pathogens of plants, animals and humans[87] that could instead undermine the aim to grow the global economy.

## Methods

*Spodoptera frugiperda* populations sampled and analysed in this study were sourced from Florida (*n* = 24)[79], Mississippi (Stoneville; *n* = 18)[30], Puerto Rico (Ayala; *n* = 15)[88], Peru (*n* = 16), Brazil (*n* = 12; IBAMA Permit number: 18BR028445/DF), Mexico (Texcoco, Estado de Mexico, sampling date: 2009; *n* = 10), Guadeloupe (*n* = 4), French Guiana (*n* = 3), Benin (*n* = 4), India (*n* = 12)[89], Tanzania (*n* = 1), Uganda (*n* = 15), Malawi (*n* = 16), and three populations from Yunnan Province, China (CC = 19; CY = 12; CX = 15)[40], and one individual (CH06) from Australia's pre-border interception program overseen by the Department of Agriculture, Water and the Environment (DAWE), also from Yunnan, China (Supplementary Data 1). Sampling of FAW did not require ethics approval as this was an invertebrate/insect pest widely found occurring and attacking agricultural crops. The initial differentiation of these individuals as 'corn-preferred' or 'rice-preferred' was based on the partial mtCOI gene region[41] and a polymorphism within the Triose Phosphate Isomerase (*Tpi*) gene[48].

The genomes of both *Sfr* and *Sfc* have been sequenced and annotated[30], allowing higher resolution analysis of genetic structure, migration patterns and sub-species status based on a high number of genome-wide SNPs to enable identification of the potential New World origins, and the species and admixture status of the invasive *Sfc* and *Sfr* populations. Extraction of total genomic DNA was carried out at the CSIRO Black Mountain Laboratories site in Canberra Australia for the Brazil, Tanzania, Malawi and Uganda populations, as well as the pre-border intercepted FAW sample from Peru and China, using the Qiagen Blood and Tissue DNA extraction kit following instructions as provided, with genomic DNA eluted in 200 μL EB. Total genomic DNA for the other three Chinese populations were extracted at Nanjing Agricultural University as detailed in Guan et al.[40]. Total genomic DNA from Mississippi, Florida, Puerto Rico, Guadeloupe, Mexico, and French Guiana, and Indian populations was carried out at INRAE DGIMI (Univ. Montpellier, INRAE, France) as reported in Yainna et al.[89].

Genomic libraries prepared by CSIRO were constructed using an Illumina Nextera Flex DNA Library Prep Kit following manufacturer's instructions and sequenced by Illumina NovaSeq6000 S4 300 sequencing system at the Australian Genome Research Facility (AGRF). Sequencing efforts were shared between three research institutions: 61 samples were prepared at CSIRO (populations from Brazil, Peru, Uganda, Tanzania, and Malawi), 46 samples were prepared by NJAU for populations from China Yunnan Province (CC, CY and CX counties), and 89 samples were prepared by DGIMI, France (populations from Florida, Mississippi, Puerto Rico, Guadeloupe, French Guiana, Mexico, Benin and India). The Peru FAW samples and the single FAW sample CH06 from Yunnan China were intercepted at Australia's pre-border inspections of imported agricultural and horticultural commodities by the Department of Agriculture, Water and the Environment (DAWE) on fresh vegetables and cut flowers, respectively. The FAW CH06 was sequenced using the Illumina MiSeq high throughput sequencing (HTS) platform following the methods of Tay et al.[90]. Sequencing coverage ranged from 2–56× with a mean coverage of 19×.

**Mitochondrial genomes assembly and haplotypes characterisation.** The mitochondrial DNA genome for all samples were assembled using Geneious 11.1.5 based on strategies previously used for assembly of *Helicoverpa* species as outlined in Walsh et al.[91]. Assembled mitogenomes were annotated using MITOS[92] selecting invertebrate mitochondrial genetic code. All annotated protein-coding genes/coding sequences (PCGs/CDS) were re-annotated visually to identify putative stop codons and to align start codon positions. Four regions of low complexity (corresponding to BC55 nt6065–6092; nt9544–9580; nt12807–12838; nt15047–15276) were trimmed due to alignment difficulties and low genome assembly confidence associated with simple repeat units, resulting in all samples having final mitochondrial DNA genome length of 15,059 bp. We identified unique mitogenome haplotypes using the DNAcollapser in FaBox (1.5) <https://users-birc.au.dk/~palle/php/fabox/dnacollapser.php>[93] after alignment using MAFFT Alignment v7.450[94,95] within Geneious 11.1.5 and selecting the Auto option for Algorithm and 200PAM / K = 2 for Scoring matrix, Gap open penalty of 1.53, and offset value of 0.123. GenBank accession numbers for full mitochondrial genomes from all individuals are listed in Supplementary Data 1.

**Nuclear SNPs selection.** In this study, we used the originally assembled genome[30] for our raw data processing. While the nuclear genomes of the two strains were found to be ~1.9% different[30], however, invasive populations analysed to-date have consisted predominantly of hybrids[31,89]. We used the native rice reference genome *Sf*R from Florida (see also[88,89] for high-quality assemblies of native population genomes for *Sfr*, and[96] for high-quality genome assemblies of native *Sfc*[31,97,98]) to map as it was found to be superior to the corn genome in terms of assembly statistics (e.g., Busco score for the corn strain indicated more missing genes than the rice strain; N50 contig size is greater for *Sfr*; see[30]). Genomic raw data was cleaned and trimmed using Trimmomatic (v0.39)[99] and aligned to the *S. frugiperda*[30] (rice v1) genome using BWA-MEM (v2.1)[100]. Variants were predicted using BBMap (v38.81)[101] using the following parameters: bgzip = t maxcov = 300 ploidy = 2 multisample = t; followed by indel normalisation using BCFtools (v1.10)[102] to obtain a whole-genome SNP panel. Variants were filtered to remove SNPs with minimum allele frequency of 0.01, any missing data and linkage disequilibrium (LD) pruned with stringent parameters (–indep-pairwise 50 kb

10.000001) using Plink2.0[103] < http://pngu.mgh.harvard.edu/purcell/plink/> to obtain 870 unlinked SNPs across all individuals.

**Phylogeny analyses**. Unrooted phylogenies based on trimmed partial mitochondrial DNA genomes of 15,059 bp and from genome-wide SNPs were individually inferred using IQ-Tree <http://iqtree.cibiv.univie.ac.at>[104]. For the nuclear SNPs, the panel of 870 SNPs from each individual in fasta format was uploaded to the IQ-Tree web server and selecting the automatic substitution model option with ascertainment bias correction (+ASC) model[105]. For the mitochondrial DNA genome maximum likelihood (ML) phylogeny was inferred with edge-linked partition for the 13 protein-coding genes and excluding all four regions of low complexity (best substitution models identified by the IQ-Tree automatic model selection option for the mitochondrial genomes as: HKY + F + I (*COI*), HKY + F + I (*COII*), TPM3u + F (*ATP8*),F81 + F (*ATP6*), HKY + F (*COIII*), HKY + F + I (*ND3*), HKY + F (*ND5*), HKY + F + I (*ND4*), HKY + F + I (*ND4L*), HKY + F + I (*ND6*), HKY + F (*CYTB*), HKY + F + I (*ND1*), HKY + F + I (*ND2*); ML phylogeny Log-likelihood: −17149.5022 ± 187.2074 s.e.). We used the Ultrafast bootstrap (UFBoot) analysis[106] with 1000 bootstrap alignments to assess branch support for both mitochondrial DNA genome and nuclear SNPs phylogenies. We implemented the default IQ-TREE settings by specifying 1000 maximum iterations and 0.99 minimum correlation coefficient, single branch tests by SH-aLRT with 1000 replications, and default IQ-TREE search parameters (perturbation strength = 0.5; IQ-TREE stopping rule: 100). Output consensus tree files in Newick format were visualised and manipulated using Dendroscope version 3.5.7[107]. We did not include an outgroup as the study was to understand inter-strain differences and not to test hypothesis relating to speciation. Due to the close evolutionary relationship between the *Sfc* and the *Sfr* (0.23–3.56 Mya;[30,66]) the inclusion of an outgroup could obscure the recent migration signals (e.g., see[108]).

**Genetic diversity and neutrality tests**. Observed ($H_{obs}$) and expected ($H_{exp}$) heterozygosity were calculated for each population using the populations program in Stacks[109] and the Adegenet package in R[110,111]. The number of loci departing significantly from Hardy-Weinberg equilibrium (HWE) in the global population and individual populations was assessed using PLINK 2.0[112] and VCFtools[113]. To test for neutrality, Tajima's D[114] and Fu and Li's D*[115] were calculated for each population using the PopGenome package in R[116]. Nucleotide diversity (π) and Wright's inbreeding coefficient, $F_{IS}$[117], were calculated using the populations program in Stacks. Pairwise comparisons of weighted $F_{ST}$ values between populations were calculated using Genepop (v4.7.5)[118] and differentiation between populations tested for significance using the exact G test.

**Population structure and migration**. Principal component analysis (PCA) was performed using PLINK v1.9[103]. The admixture was estimated using Admixture v1.3.0[119]. For the plotting of networks, the R package NetView[120,121] was used. The network drawn using the plotAdmixture function in this package is based on a maximum likelihood (ML) distance matrix calculated from the IQ-Tree phylogeny, using the R package 'ape'[122].

To estimate directional gene flow between the populations, as well as the relative magnitudes of these flows, the function divMigrate in the R package diveRsity[123] online version was used <https://popgen.shinyapps.io/divMigrate-online/>[124]. Gene flows between all sites were calculated and then normalized to obtain relative migration rates (between 0 and 1). The program divMigrate searches for gene flow directionality between each pair of populations by identifying significant asymmetry based on allele frequency, and against a hypothetically defined pool of migrants to estimate genetic differentiation between each population pair and the hypothetical pool. The relative levels of migration between the two populations are then estimated based on the resulting directional genetic differentiation[124]. To evaluate the significance of asymmetric migration, 1000 bootstraps were performed. Resulting migration matrices were then plotted using Gephi <https://gephi.org/>[125] to generate network graphs. These show directional gene flows between populations (located at the nodes), with the thickness of the lines showing the relative strength of gene flow.

**Statistics and reproducibility**. Experimental design and statistical details performed in this study are provided in the respective "Results" and "Methods" sections. Sample size of populations are provided in the "Methods" section. All programs were run with default parameters and statistics are absolute estimates with no replicates or corrections required for either.

**Reporting summary**. Further information on research design is available in the Nature Research Reporting Summary linked to this article.

## Data availability

All assembled mitochondrial genomes have been submitted to GenBank (accession numbers MT897262 - MT897458). The complete list of FAW population genome wide single nuclear polymorphic loci used is available from CSIRO's public data access portal https://data.csiro.au/collection/csiro:53315, https://doi.org/10.25919/y3nd-2903[126].

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

## Acknowledgements

This work was funded by CSIRO Health & Biosecurity (R-91044-01) to W.T.T., T.W., K.H.J.G, S.E., and S.D. W.T.T., T.W., S.E., K.H.J., and D.K. were also funded by CSIRO Health & Biosecurity 'Genes of Biosecurity importance fund' (R-8681-1), and D.K. also acknowledged funding by CSIRO (R-90035-14). R.R. was funded by CSIRO and Hort Innovation Australia (HIA). A.P. was supported by CSIRO, CRDC and Bayer (R10801-01). J.Z. and Y.W. were supported by the National Key Research Development Program of China (No. 2019YFD0300103 to Y.W.) and the Fundamental Research Funds for the Central Universities of China (KYZ201920 to Y.W.). M.H.O. was funded by the Agricultural Technology and Agribusiness Advisory Services Project (Project ID No.: P109224) funded by the World Bank. The work performed at INRAE was publicly funded through the ANR (the French National Research Agency, Grant ID 1702-018, given to K.N.) under the "Investissements d'avenir" programme with the reference ANR-10-LABX-001-01 Labex Agro and coordinated by Agropolis Fondation under the frame of I-SITE MUSE (ANR-16-IDEX-0006). It was also funded by a grant from the department of Santé des Plantes et Environnement at Institut national de la recherche agronomique for K.N. (adaptivesv). It was also financially supported by EUPHRESCO (FAW-spedcom, given to Anne-Nathalie Volkoff). We thank Stella Adumo (NaCRRI) with Uganda FAW sample collection, and Carlos Blanco for the Mexican FAW samples. Peru and China FAW pre-border interception specimens were provided by the DAWE. We thank Paul De Barro and Andy Sheppard (CSIRO) for the helpful discussion. Dr Dao Thi Hang (Plant Protection Research Institute, Vietnam) assisted with sourcing and translation of reference material from Vietnam National University of Agriculture. Dr Nguyen Van Liem, Dr Dao Thi Hang (Plant Protection Research Institute, Vietnam), and Prof. Dr. Vu Van Lien (Deputy General Director, Vietnam National Museum of Nature, VAST) for helpful discussion.

## Author contributions

W.T.T., T.K.W., K.N., E.d'A., N.N., S.D., D.J.K., C.C., and K.H.J.G. designed the study. W.T.T., K.H.J.G., T.W., R.V.R., A.P., S.E., D.K., Z.J., and Y.W. analysed the data. W.T.T., T.W., S.D., Y.W., K.N., E.d'A., N.N., C.C., and M.H.O. sourced the samples. W.T.T., T.K.W., K.H.J.G., R.V.R., and D.J.K. drafted the manuscript, all authors contributed on improving the manuscript. W.T.T., R.V.R., K.H.J.G., K.N., T.K.W., N.N., and M.H.O. revised the manuscript.

## Competing interests

The authors declare no competing interests.
