## [Peer Review File · Communications Biology]

Reviewers' comments:

Reviewer #1 (Remarks to the Author):

Comments to Author:

The authors analyzed patterns of variation among 197 *Spodoptera frugiperda* (fall armyworm, FAW) genomes, representing a worldwide sample from both Old World and New World. They studied the global FAW movement history based on genomic data, supporting a multiple introduction model of the FAW invasion from the Americas into the Old World. Thereby it adds to a broader understanding of FAW's global spread, and it's also important for the management of FAW. It could be a potential high-quality paper, but the following concerns should be addressed.

Major comments:

1. The descriptions of methods are very unclear and hard to follow. More details about genome sequencing (including sequencing coverage, library construction), data filtering, mapping, SNP calling and construction of the phylogenetic trees are needed. In addition, some important parameter settings and software version are all missing.
2. Page4, line17. How different at the genomic level between the rice-preferred and corn-preferred FAW? Author only aligned sequencing reads to the genome of rice-preferred FAW according to their methods. It may cause problems if there are some major differences between the genomes of these two FAW strains. Need some explanations about the reason why you used rice-preferred FAW as reference genome.
3. Page5, line 4. Please explain why you analyzed the mitochondrial genome firstly, then nuclear genome. What is the difference between them? I found the results from mitochondrial genome and nuclear genome are totally different.
4. Page5, Fig 2. You need an ideal outgroup for your phylogeny, such as a close-related species, *Spodoptera litura*. The same problem for Fig 3.
5. CH06 is a strange individual. It clustered strongly with an individual from Mississippi (UM04) and many Peru samples based on mitochondrial genome. However, using Nuclear SNPs as signal, it clustered with Florida populations. Author should explain this discrepancy.
6. The population genetics analysis provides a dense map of variants and high level of polymorphism. Additional genome-wide association study could be done to check the genomic signatures of evolutionary adaptation for FAW (for example, selections related to insecticide resistance).
7. Any difference in the movement history between rice-preferred and corn-preferred FAW strains? Different country has different kinds of crops, so the movement history of these two FAW strains may also differ, as they have different preferences.
8. I am not sure if the authors can present a clear pattern (or a figure abstract) to the audience about how FAWs spread from their native place to all over the world.

Minor issues:

1. Page5, Fig 3. There might be a mistake with the diagram annotation (China-HIA006).
2. Page4, line23. SNPs, not "SPSs".

Data availability: Could not locate raw data used in this study. could not check the GenBank link as it seems to be un-released.

Reviewer #2 (Remarks to the Author):

In “Global FAW population genomic signature supports complex introduction events across the Old World”, Tay et al. describe an analysis of population structure among global collections of the agricultural insect pest *Spodoptera frugiperda* using mitochondrial and nuclear markers and a suite of population genetics techniques. The results, if sound, would overturn the idea that the chronological sequence of recent outbreak reports in Africa and Asia reflect the true invasion routes, and also cast doubt on the validity of the current approaches to distinguishing apparent host-races (corn- and rice-preferring). I take no issue with either of these conclusions, but I am not convinced from my reading of this manuscript. This is largely because much of the methods section is difficult to assess due to a lack of adequate detail. There also appear to be some potentially worrisome issues with interpretation of population genetics results. More of a cosmetic comment: the manuscript is overly lengthy, with apparently redundant figures and long explanations of minute details of results that are very specific to the study system. Overall, I find the manuscript in need of revision, and given the narrow focus throughout on *S. frugiperda*, I think it may be a better fit in a more specific journal as well.

Disclaimer: I am not an expert on *S. frugiperda* and cannot become one within the time frame of this review request, so I cannot speak to the accuracy of anything about the *S. frugiperda* narrative and consequently the novelty of the conclusions. I instead focus my comments on the clarity of the presentation of results and interpretations of population genetics analyses.

Lines 13-25: This is excellent background. However, it is still unclear how this spread is occurring – what aspect of global trade, and what life stage, is facilitating FAW spread? E.g. are pupae being moved in soil, larvae being moved in fresh vegetables?

Page 4 Line 10. It is unclear what kinds of sequences these SNPs were derived from. Whole genomes? The mitochondrial genomes described just above?

Page 4 Line 20. I am having a hard time digesting how whole-genome data could be widdled down to 870 SNPs. Please clarify.

Page 4 Line 30. I assume these analyses were done with the nuclear SNPs? Please clarify.

Page 4 Line 44-51. I am unfamiliar with this method, and after reading the explanation, still do not know how it infers directionality. Please clarify.

Figure 2. Adding a legend to the figure that also explains the meanings of the line colors would be nice.

Page 7 Line 46-48. Authors do not suggest the common interpretation of heterozygote excess, which is that a population recently underwent a bottleneck (Maruyama 1985), which would naturally be expected among samples taken from invasive populations.

Maruyama, T., Fuerst, P.A. Population bottlenecks and nonequilibrium models in population genetics. II. Number of alleles in a small population that was formed by a recent bottleneck. *Genetics* 1985, 111, 675–689.

Page 8 Line 3 “and highlighted the complex global population structure” is vague and does not follow from the results presented in this section.

Page 8 Line 5. Here again, the common interpretation is ignored. Positive Tajima’s D is usually taken as evidence of either balancing selection or a population bottleneck. In Line 13-14 the authors suggest a bottleneck was unlikely to affect genomic variation within the time frames in which the populations were sampled. This makes no sense. If a bottleneck occurred upon introduction within the last decade, we can certainly expect to detect its signature in the genome – marked by an underrepresentation of low frequency alleles.

Table 1. These are very high values of nucleotide diversity. It is unclear what data were used to generate these statistics, but I would hazard a guess that only a handful of highly polymorphic loci

were used?

How do Figure 3 and Figure 2 differ? I assume Figure 3 relies on nuclear SNP data? I like the color scheme in Figure 3, and wonder if it couldn't also be used for Figure 2?

What do the red dots at some nodes in Figure 3 mean?

Page 10 Line 7. F_{ST} does not equate to a measure of gene flow – multiple evolutionary processes can be relevant to measures of genetic differentiation (like F_{ST}).

Figure 4. I think I understand what the authors were doing, but showing the same plot with various groups shown or not shown can be confusing, especially when only a subset of the points shown are referred to in the respective figure legends.

Figure 5. Why were only $K=3$ to $K=5$ considered? What do the red dots mean?

Figures 6a and 6b. This is an odd way to number figures. I suggest moving one to the supplement and calling the other simply Figure 6. These figures seem redundant with the phylogenetic trees presented in Figures 2-3, but they are much more difficult to read given the large number of colors and abbreviations throughout.

Page 19 Line 3-5: This reminds me of a “bridgehead” effect that has been suggested as a general feature of global invertebrate invasions, something that the authors might use to help frame their discussion (I confirm no conflict of interest in suggesting this citation). Guillemaud, T., Ciosi, M., Lombaert, É., & Estoup, A. (2011). Biological invasions in agricultural settings: Insights from evolutionary biology and population genetics. *Comptes Rendus Biologies*, 334(3), 237–246. <https://doi.org/10.1016/j.crvi.2010.12.008>

Reviewer #3 (Remarks to the Author):

Instead of attaching my review in a separate document I had decided to type directly into the review comments page, which was apparently a mistake. When submitting the review, the site stated that my login has expired and cleared the form along with all of my text that I should have saved elsewhere. I apologize, but I don't have time to repeat the more detailed review here again.

In general I find the study to be excellent with no major problems. It would be interesting to know the exact location where the Mexican specimens were collected as that would indicate where the findings are unusual or not. It would be good if some of the figures were clearer, and I especially do not like the circular tree in Fig. 3, although I understand that is done to save space. Otherwise I generally agree with everything the authors conclude and recommend publication.

Reviewers' comments:

Reviewer #1 (Remarks to the Author):

Comments to Author:

The authors analyzed patterns of variation among 197 *Spodoptera frugiperda* (fall armyworm, FAW) genomes, representing a worldwide sample from both Old World and New World. They studied the global FAW movement history based on genomic data, supporting a multiple introduction model of the FAW invasion from the Americas into the Old World. Thereby it adds to a broader understanding of FAW's global spread, and it's also important for the management of FAW. It could be a potential high-quality paper, but the following concerns should be addressed.

Major comments:

1. The descriptions of methods are very unclear and hard to follow. More details about genome sequencing (including sequencing coverage, library construction), data filtering, mapping, SNP calling and construction of the phylogenetic trees are needed. In addition, some important parameter settings and software version are all missing.

[Authors]: We have now added further details to the methods section including programs' version, citations, and parameters used for Nuclear SNPs selection and for phylogenetic tree reconstruction (P4L7 – L39).

2. Page4, line17. How different at the genomic level between the rice-preferred and corn-preferred FAW? Author only aligned sequencing reads to the genome of rice-preferred FAW according to their methods. It may cause problems if there are some major differences between the genomes of these two FAW strains. Need some explanations about the reason why you used rice-preferred FAW as reference genome.

[Authors]: The nuclear genomes of the two strains are found to be ~1.9% different (P2L34; P4L9), however, the invasive populations are predominantly hybrids (e.g., Zhang et al. 2020 ME; Yainna et al. 2020; Gui et al. 2020) (P4L5-10) and we used the rice genome to map as it was found to be superior to the corn genome in terms of assembly statistics (e.g., Busco score for the corn strain indicated more missing genes than the rice strain; N50 contig size is greater for Sfr; see Gouin et al. 2017) (P4L10-14). We have added this to the material and methods section and we thank the R2 for the suggestion.

3. Page5, line 4. Please explain why you analyzed the mitochondrial genome firstly, then nuclear genome. What is the difference between them ? I found the results from mitochondrial genome and nuclear genome are totally different.

[Authors]: We analysed the mitochondrial genomes first because this is where the widely used partial mtCOI gene resides, which is one of the main markers used by large portions of research community to differentiate between the corn- and rice-preferred FAW. The results from the mitochondrial genome and nuclear genome are expected to be different because of their different modes of inheritance – with the mitochondrial genome being (almost) exclusively maternally inherited and lacking recombination, while the nuclear genome being biparental and with recombination. We have now up-dated the relevant Results section as requested by R1 (P5L16-23).

As found in our analysis, and also the studies by, e.g., Zhang et al. (2020); Yainna et al. (2020), the majority of the invasive FAW individuals from populations surveyed thus far all exhibited admixed genome signatures, suggesting that they were hybrids of corn- and rice-preferred FAW. The mitochondrial genome would only show the maternal 'rice' or 'corn' lineage, and not reflect the potential 'hybrid' origins of these individuals. Results between mitochondrial and nuclear genomes are therefore not expected to be the same.

4. Page5, Fig 2. You need an ideal outgroup for your phylogeny, such as a close-related species, *Spodoptera litura*. The same problem for Fig 3.

[Authors]: We thank the reviewer for the suggestion to include an outgroup for our analysis. However, our aim of the study was not to generate a rooted phylogeny to infer directionality of character change, nor were we trying to understand the evolution of specific traits along our phylogeny. We have now clarified this by stating in the revised manuscript that our phylogenies are unrooted. Our study was to compare the genomic signatures relating to invasion biology between these two highly related sister clades of FAW, and the inclusion of an outgroup (e.g., see Gui et al. 2020) would significantly reduce the resolution to obscure their recent population history. Furthermore, as demonstrated by Gui et al. (2020), inclusion of an outgroup that is much more divergent than the target species would reduce our ability to detect evolutionary signals at the population level and reduced admixture signal, as well as leading to incorrect inference (e.g., one does not include a primate outgroup in human population genetics study; see Hellenthal et al. 2014). This is now clarified in P4L36-39.

We also note that since the evolutionary timeframe between the ‘corn’ and ‘rice’ FAW was short (~2 mya; Gouin et al. 2017; Arias et al. 2019); the use of a distantly related species such as *S. litura* (see kergoat et al. 2021) would complicate interpretations. Based on the study of Kergoat et al. (2021), a potential outgroup species for a corn/rice FAW rooted phylogeny would be *S. apertura*, for which there is currently no whole genome sequence data available.

5. CH06 is a strange individual. It clustered strongly with an individual from Mississippi (UM04) and many Peru samples based on mitochondrial genome. However, using Nuclear SNPs as signal, it clustered with Florida populations. Author should explain this discrepancy.

[Authors]: We concur that CH06’s phylogenetic position to be in conflict depending on whether inference was from mitochondrial or nuclear genome. As detailed above (see #3), this likely reflects the non-recombination characteristics of the mitochondrial genome and the potential admixed nature of nuclear genome. We note that within this clade (node support 80%) it included also various individuals from across the North and South Americas, including Florida (e.g., UF01, UF11), Brazil (BR20, BC61, BC55), and Mississippi (UM05, UM02), in addition to the Peruvian individuals P5L9-10, and likely reflect the long-distance migratory nature of this noctuid pest (see Fig. 2). In the study by Gui et al. (2020), the authors demonstrated a Yunnan individual (NJ05) was clustered also with US (‘America B’ from Louisiana) individuals, while as a whole, Louisiana (‘America C’) and Florida (‘America D’) were basal sister clades to the Yunnan and Guangxi populations analysed by genome-wide SNPs. We have amended the relevant sections accordingly P18L11-12; P18L30-31. We thank R1 for his/her suggestion.

6. The population genetics analysis provides a dense map of variants and high level of polymorphism. Additional genome-wide association study could be done to check the genomic signatures of evolutionary adaptation for FAW (for example, selections related to insecticide resistance).

[Authors]: Our aim was to disentangle the genomic signatures relating to introduction of this highly invasive pest complex and to test the alternative hypothesis of multiple introductions against the axiom of rapid west-to-east spread as the result of a single introduction. Characterisation of known resistance genes in these populations including with respect to e.g., pyrethroid, organophosphate, diamides, and Bt resistances, were separately reported by Guan et al. (2021) and Yainna et al. (2021).

7. Any difference in the movement history between rice-preferred and corn-preferred FAW strains? Different country has different kinds of crops, so the movement history of these two FAW strains may also different, as they have different preferences.

[Authors]: Our genome-wide SNP analyses of these invasive populations showed admixed genome in the majority of individuals. The invasive populations should therefore best be regarded as ‘hybrids’. Our admixture analysis showed that by and large, there were no ‘pure’ corn/rice FAW and attempts to understand movement differences as suggested by R1 in these invasive populations would therefore unlikely to be successful. We note that the persistence to classify the invasive populations as either rice- or corn-FAW despite their hybrid status as shown by various genomic analyses (e.g., Zhang et al. 2020; Gui et al. 2020; this study) will not help with ‘predicting’ which crops will likely be attacked. We have added the appropriate sentence to the Discussion section **P18L32-34**.

8. I am not sure if the authors can present a clear pattern (or a figure abstract) to the audience about how FAWs spread from their native place to all over the world.

[Authors]: FAW is likely a pest that had been transported from their native geographical range to various locations around the world due to increased movements of associated agricultural commodities and potentially over substantially long period of time. For example, the FAW infestation was already reported in Israel/Jordan Valley in 1967 (Wiltshire 1977), in Germany in 1999 (Jeger et al. 2017) although this population did not persist due to the extreme cold winter weather in central/western Europe, in Vietnam since 2008 (Vu 2008; Nguyen and Vu 2009); and in intercepted cut flowers that originated from Yunan since 2016 (Tay and Gordon 2019; this study for CH006 sample), the year when the pest was supposedly first detected in western Africa (Goergen et al. 2016). US pre-border interceptions of suspect larvae identified as FAW by the Identification Technology Program (ITP) from other non-American regions have also been reported since before 2014 (see Gilligan and Passoa 2014; <<https://idtools.org/id/leps/lepintercept/frugiperda.html>>, accessed 21-July, 2021) from countries such as Israel, Turkey, China, Indonesia, Thailand, Micronesia, and the Netherlands (which likely included suspected individuals that originated from various African and Asian countries with cut flower industries).

We appreciate this suggestion by R1 and indeed it would be ideal to be able to present a figure abstract with regards how FAW potentially spread from their native place, however with significant knowledge gap for populations in e.g., the near East, Southeast Asia, and other Asian regions (e.g., populations from other Chinese and Indian locations), it would be difficult and premature to present a clear picture of potential spread for this pest.

Minor issues:

1. Page5, Fig 3. There might be a mistake with the diagram annotation (China-HIA006).

[Authors]: Fixed (**see up-dated Figure 2 below**)

2. Page4, line23. SNPs, not “SPSs”.

[Authors]: Fixed

Data availability: Could not locate raw data used in this study. could not check the GenBank link as it seems to be un-released.

[Authors]: Raw data have now been up-loaded to the CSIRO data repository and are publicly downloadable. The link has been provided in “Statement on Data Availability” section. We have also instructed NCBI to release all GenBank accessions for mitochondrial genomes from this study.

Reviewer #2 (Remarks to the Author):

In “Global FAW population genomic signature supports complex introduction events across the Old World”, Tay et al. describe an analysis of population structure among global collections of the agricultural insect pest *Spodoptera frugiperda* using mitochondrial and nuclear markers and a suite of population genetics techniques. The results, if sound, would overturn the idea that the chronological sequence of recent outbreak reports in Africa and Asia reflect the true invasion routes, and also cast doubt on the validity of the current approaches to distinguishing apparent host-races (corn- and rice-preferring). I take no issue with either of these conclusions, but I am not convinced from my reading of this manuscript. This is largely because much of the methods section is difficult to assess due to a lack of adequate detail. There also appear to be some potentially worrisome issues with interpretation of population genetics results. More of a cosmetic comment: the manuscript is overly lengthy, with apparently redundant figures and long explanations of minute details of results that are very specific to the study system. Overall, I find the manuscript in need of revision, and given the narrow focus throughout on *S. frugiperda*, I think it may be a better fit in a more specific journal as well.

Disclaimer: I am not an expert on *S. frugiperda* and cannot become one within the time frame of this review request, so I cannot speak to the accuracy of anything about the *S. frugiperda* narrative and consequently the novelty of the conclusions. I instead focus my comments on the clarity of the presentation of results and interpretations of population genetics analyses.

Lines 13-25: This is excellent background. However, it is still unclear how this spread is occurring – what aspect of global trade, and what life stage, is facilitating FAW spread? E.g. are pupae being moved in soil, larvae being moved in fresh vegetables?

[Authors]: We have now provided further details on the aspects of global trade, and life stages, that are facilitating the spread of this pest. See P2L16-22.

Page 4 Line 10. It is unclear what kinds of sequences these SNPs were derived from. Whole genomes? The mitochondrial genomes described just above?

[Authors]: The SNPs were derived from whole genome sequencing see P4L7-20.

Page 4 Line 20. I am having a hard time digesting how whole-genome data could be widdled down to 870 SNPs. Please clarify.

[Authors]: To accurately infer population structure, we required neutral and completely unlinked loci. The approach we used to obtain the same is as below:

The whole genome sequencing data from all samples were normalised and filtered to remove missing data. This was followed by linkage disequilibrium (LD) based pruning with stringent parameters (--indep-pairwise 50kb 1 0.000001). The r^2 threshold (0.000001) was used to obtain sites that are strictly genetically independent of each other. Conventional LD pruning uses an r^2 threshold of 0.1 or 0.01, which is much more relaxed and can maintain SNP's that are in closer vicinity to each other. We have provided details in the revised Nuclear SNPs selection section see P4L7-20.

Page 4 Line 30. I assume these analyses were done with the nuclear SNPs? Please clarify.

[Authors]: We have clarified that these were nuclear SNPs see P4L7-20.

Page 4 Line 44-51. I am unfamiliar with this method, and after reading the explanation, still do not know how it infers directionality. Please clarify.

[Authors]: DivMigrate uses the asymmetric nature of gene flow and divergence estimates to calculate migration patterns from populations, which are in turn used to parsimoniously estimate directionality. We have modified the relevant section P5L7-10 to better explain the concept, and also provided the relevant citation Sundqvist et al. 2016, (in text citation #99), see P5L10 for further reading.

Figure 2. Adding a legend to the figure that also explains the meanings of the line colors would be nice.

[Authors]: Fixed, a legend has been added, please see up-dated Fig. 2 below. We have also changed: (i) the branch colours to denote only their rice-/corn-preference based on mtCOI, (ii) for the invasive lineages originally indicated by red branches, we have now changed to indicate these by red dots; and (iii) we now only showed bootstrap support values of $\geq 50\%$ to simplify the figure.

Fig. 2 up-dated:

Page 7 Line 46-48. Authors do not suggest the common interpretation of heterozygote excess, which is that a population recently underwent a bottleneck (Maruyama 1985), which would naturally be expected among samples taken from invasive populations.

Maruyama, T., Fuerst, P.A. Population bottlenecks and nonequilibrium models in population genetics. II. Number of alleles in a small population that was formed by a recent bottleneck. *Genetics* 1985, 111, 675–689.

[Authors]: We thank the reviewer for his/her comment regarding interpretation of heterozygote excess. We have now significantly improved the relevant sections (P81,25-35), clarifying that the excess in heterozygosity was between the observed and the expected heterozygosity levels (i.e., $H_{exp} < H_{obs}$; see Luikart & Cornuet 1998) in these invasive populations, which therefore did not support these populations to have undergone bottleneck in recent times (as would be expected from samples taken from invasive populations noted by R2).

Page 8 Line 3 “and highlighted the complex global population structure” is vague and does not follow from the results presented in this section.

[Authors]: Fixed. We have further elaborated on the statement to consider the complex migratory behaviour of *S. frugiperda* and associated implications in both native (especially in Central and South Americas) and the current invasive ranges (P8L41-46).

Page 8 Line 5. Here again, the common interpretation is ignored. Positive Tajima’s D is usually taken as evidence of either balancing selection or a population bottleneck. In Line 13-14 the authors suggest a bottleneck was unlikely to affect genomic variation within the time frames in which the populations were sampled. This makes no sense. If a bottleneck occurred upon introduction within the last decade, we can certainly expect to detect its signature in the genome – marked by an underrepresentation of low frequency alleles.

[Authors]: We have significantly reworked this section including better explanation and interpretation of Tajima’s D (e.g., positive Tajima’s D signifies balancing selection/population reduction/population substructure/recent bottleneck leading to a decrease in allelic diversity (i.e., due to a lack of rare alleles) as compared with observed heterozygosity), as well as removing repetitive (and somewhat confusing) sentences (e.g., as pointed out by R2 for L13-14). Our analyses of Tajima’s D, and Fu and Li’s D* especially for invasive populations were consistent with these populations being established from recent bottleneck events and/or experienced balancing selection, rather than from a western African ‘bridgehead effect’ leading to subsequent geographic population expansion as the pest rapidly spread from eastern Africa to Asia.

We also note that neutrality test statistics such as Tajima’s D and Fu-Li’s D* are affected by sample sizes as demonstrated by Subramanian (2016) and our estimates appeared to show this trend. We have therefore modified Table 1 by transferring the columns on Tajima’s D and Fu-Li’s D* statistics, as well as the relevant sections in the main text, to Supplemental Table 2, and clearly indicated interpretations of the finding should proceed with caution, as well as referring readers to the study of Subramanian (2016).

Table 1. These are very high values of nucleotide diversity. It is unclear what data were used to generate these statistics, but I would hazard a guess that only a handful of highly polymorphic loci were used?

[Authors]: Fixed. R2’s assumption was correct. We clearly stated that these were variable independent and included no invariant loci (P8L23-24) in the Results section, and we have also now provided a statement in the Table note to clarify that the high nucleotide diversity estimates were from low number (i.e., 870 SNPs, in relation to whole genome SNPs) of neutral unlinked polymorphic markers, and hence of comparative value within this study and future studies that utilised similar set of SNP markers (P9L8-10).

How do Figure 3 and Figure 2 differ? I assume Figure 3 relies on nuclear SNP data? I like the color scheme in Figure 3, and wonder if it couldn’t also be used for Figure 2?

[Authors]: Figure 2 was from mitochondrial DNA genomes and included only unique genomes (i.e., non-redundant). Fig. 3 was from all individuals and based on the selected 870 SNPs. We have modified Fig. 2 as suggested by R1.

What do the red dots at some nodes in Figure 3 mean?

[Authors]: This was clearly stated in the Fig 3 caption, i.e., ‘Branch nodes with 100% bootstrap support are indicated by red dots’ (P10L8-9).

Page 10 Line 7. FST does not equate to a measure of gene flow – multiple evolutionary processes can be relevant to measures of genetic differentiation (like FST).

[Authors]: Due to the difficulty of directly measuring gene flow, we adopted the F_{ST} (a measure of genetic variance among populations) as an indirect measure of gene flow (see Whitlock and McCauley 1999). We appreciate R2's comments and to avoid potential confusion, we have reworded the relevant paragraphs [P11L6-8, P11L12, P12L1, P18L41-42] by clarifying that it was population genetic differentiation (F_{ST}) that was estimated to indirectly infer gene flow between populations.

Figure 4. I think I understand what the authors were doing, but showing the same plot with various groups shown or not shown can be confusing, especially when only a subset of the points shown are referred to in the respective figure legends.

[Authors]: We appreciate R2's comments. Due to the complexity of the datapoints, we feel that it would make more sense to show the various native groups, and we also note that both R1 and R3 did not raise this as an issue.

Figure 5. Why were only K=3 to K=5 considered? What do the red dots mean?

[Authors]: We presented only K=3 to K=5 because at these K numbers their respective CV error values were the lowest. We have not presented more as we did not wish to lengthen the manuscript unnecessarily.

[Authors]: The red dots on the admixture graphs represent individuals that lacked signatures of admixture. We have clarified this in the figure caption [P14L2-3]. We have also changed the grey dots in panels with K=3 and K=4 to red dots, as they also represent individuals that lacked evidence of admixture. The new up-dated figure is shown:

Fig. 4 up-dated:

Figures 6a and 6b. This is an odd way to number figures. I suggest moving one to the supplement and calling the other simply Figure 6. These figures seem redundant with the phylogenetic trees presented in Figures 2-3, but they are much more difficult to read given the large number of colors and abbreviations throughout.

[Authors]: we have moved Fig. 6a to Supplemental Figure 2. We presented these figure as they represented an alternative view to show both clustering of populations and the degree of admixture in each population While we concur that individual codes were somewhat crowded however the main message was clearly presented by the country/genetic cluster colours that showed, e.g., separation between China and East African nations, to further support multiple introductions of the pest across its invasive ranges.

Page 19 Line 3-5: This reminds me of a “bridgehead” effect that has been suggested as a general feature of global invertebrate invasions, something that the authors might use to help frame their discussion (I confirm no conflict of interest in suggesting this citation). Guillemaud, T., Ciosi, M., Lombaert, É., & Estoup, A. (2011). Biological invasions in agricultural settings: Insights from evolutionary biology and population genetics. *Comptes Rendus Biologies*, 334(3), 237–246. <https://doi.org/10.1016/j.crvi.2010.12.008>

[Authors]: we thank R2 for this useful reference and we have modified relevant sections (Introduction P2L46, P2L52-P3L2); Discussion P17L15-18) and provided the citation in the revised version.

Reviewer #3 (Remarks to the Author):

Instead of attaching my review in a separate document I had decided to type directly into the review comments page, which was apparently a mistake. When submitting the review, the site stated that my login has expired and cleared the form along with all of my text that I should have saved elsewhere. I apologize, but I don't have time to repeat the more detailed review here again.

In general I find the study to be excellent with no major problems. It would be interesting to know the exact location where the Mexican specimens were collected as that would indicate where the findings are unusual or not. It would be good if some of the figures were clearer, and I especially do not like the circular tree in Fig. 3, although I understand that is done to save space. Otherwise I generally agree with everything the authors conclude and recommend publication.

[Authors]: we thank R3 for his/her comments and review of our study. R3 is correct in his/her assessment that the circular tree in Fig. 3 (but also Fig. 2) was presented with space-saving consideration. We have also provided the location and sampling year for the Mexican samples P3L16 although regretfully we do not have further information relating to this population since it was a donation by the collector (see P19L31) to the INRAE co-authors in 2009.

References

- Arias, O. *et al.* Population genetic structure and demographic history of *Spodoptera frugiperda* (Lepidoptera: Noctuidae): implications for insect resistance management programs. *Pest Management Science* **75**, 2948–2957, doi:10.1002/ps.5407 (2019).
- Gilligan, T. M. & Passoa, S., C. *LepIntercept, An identification resource for intercepted Lepidoptera larvae. Identification Technology Program (ITP)*, <<http://idtools.org/id/leps/lepintercept/frugiperda.html>> (2014).
- Goergen, G., Kumar, P. L., Sankung, S. B., Togola, A. & Tamo, M. First Report of Outbreaks of the Fall Armyworm *Spodoptera frugiperda* (J E Smith) (Lepidoptera, Noctuidae), a New Alien Invasive Pest in West and Central Africa. *PLoS One* **11**, e0165632, doi:10.1371/journal.pone.0165632 (2016).
- Gouin, A., Bretaudeau, A., Nam, K. *et al.* Two genomes of highly polyphagous lepidopteran pests (*Spodoptera frugiperda*, Noctuidae) with different host-plant ranges. *Sci Rep* **7**, 11816 (2017). <https://doi.org/10.1038/s41598-017-10461-4>
- Guan F, Zhang J, Shen H, *et al.* (2021) Whole-genome sequencing to detect mutations associated with resistance to insecticides and Bt proteins in *Spodoptera frugiperda*. *Insect Science* **28**(3), 627–638
- Gui F, *et al.* Genomic and transcriptomic analysis unveils population evolution and development of pesticide resistance in fall armyworm *Spodoptera frugiperda*. *Protein Cell*. 2020 Oct 27. doi: 10.1007/s13238-020-00795-7. Epub ahead of print. PMID: 33108584.

- Jeger, M. et al. Pest categorisation of *Spodoptera frugiperda*. *Efsa Journal* **15**, doi:<https://doi.org/10.2903/j.efsa.2017.4927> (2017).
- Kergoat GJ, et al. A novel reference dated phylogeny for the genus *Spodoptera* Guenée (Lepidoptera: Noctuidae: Noctuinae): new insights into the evolution of a pest-rich genus. *Mol Phylogenet Evol.* 2021 Aug;161:107161. doi: 10.1016/j.ympev.2021.107161. Epub 2021 Mar 29. PMID: 33794395.
- Luikart, G. & Cornuet, J. M. Empirical evaluation of a test for identifying recently bottlenecked populations from allele frequency data. *Conserv Biol* **12**, 228-237, doi:DOI 10.1046/j.1523-1739.1998.96388.x (1998).
- Nguyen, T. K. O. & Vu, T. P. Checklist of turfgrass insect pests, morphology, biology and population fluctuation of *Herpetogramma phaeopteralis* (Guenee) (Lepidoptera: Pyralidae) in Ha Noi, in spring-summer 2008. in *The 3rd National Conference of Ecology and Natural Resources*. pp. 1490-1498.
- Subramanian S. The effect of sample size on population genomic analyses - implications for the tests of neutrality. *BMC Genomics* **17**, 123 (2016).
- Sundqvist, L., Keenan, K., Zackrisson, M., Prodohl, P. & Kleinhaus, D. Directional genetic differentiation and relative migration. *Ecol Evol* **6**, 3461-3475, doi:10.1002/ece3.2096 (2016)
- Tay WT, Gordon KHJ (2019) Going global - genomic insights into insect invasions. *Current Opinion in Insect Science* **31**, 123-130.
- Vu, T. P. *Insect pests of turf grass, biology, ecology and the control of Herpetogramma phaeopteralis (Guenée) in Hà Nội in Spring Summer 2008* MSc thesis, Hà Nội Agriculture University, Vietnam, (2008).
- Whitlock MC, McCauley DE (1999) Indirect measures of gene flow and migration: $F_{ST} \neq 1/(4NM+1)$. *Heredity* **82**, 117-125.
- Wiltshire EP (1977) Middle East Lepidoptera, XXXVII: Notes on the *Spodoptera litura* (F.) - Group (Noctuidae - Trifinae). *Proc. Brit. Ent. Nat. Hist. Soc.* **10**, 92-96
- Yainna S, et al. Genomic balancing selection is key to the invasive success of the fall armyworm. *bioRxiv* 2020.06.17.154880; doi: <https://doi.org/10.1101/2020.06.17.154880> (2020)
- Yainna S. et al. Geographic Monitoring of Insecticide Resistance Mutations in Native and Invasive Populations of the Fall Armyworm. *Insects* **2021**, *12*, 468. <https://doi.org/10.3390/insects12050468>
- Zhang L, et al. Genetic structure and insecticide resistance characteristics of fall armyworm populations invading China. *Mol Ecol Resour.* 2020 Nov;20(6):1682-1696. doi: 10.1111/1755-0998.13219. Epub 2020 Jul 20. PMID: 32619331; PMCID: PMC7689805

REVIEWERS' COMMENTS:

Reviewer #3 (Remarks to the Author):

This will be a very important publication for researchers (including myself) working on FAW. The revised version of the manuscript is much improved.

The only specific comment I have is regarding the LepIntercept data, cited here (line 548):

"the pre-border interceptions of FAW that originated from countries outside of the Americas since prior to 2014..." [Citation for 112]

While we certainly appreciate the citation for LepIntercept, it is most likely that many of these records from outside the Americas up to 2014 were simply misidentifications. Many of the Spodoptera that are intercepted are early instar, and difficult to identify to species. So it is not known why these particular interceptions were identified to the species level, and they were not verified with any other methods (sequencing, etc.). So it is possible they were identified correctly, but it is much more likely that most are simply misidentifications, and at this point we haven't gone back to sequence any of this material. So I think it would be good to add more ambiguity to the above statement if possible to let the reader know these IDs are not confirmed.

Otherwise, I see no issues with the revised manuscript. Thank you for the opportunity to review!

REVIEWERS' COMMENTS:

Reviewer #3 (Remarks to the Author): This will be a very important publication for researchers (including myself) working on FAW. The revised version of the manuscript is much improved. The only specific comment I have is regarding the LepIntercept data, cited here (line 548): "the pre-border interceptions of FAW that originated from countries outside of the Americas since prior to 2014..." [Citation for 112]. While we certainly appreciate the citation for LepIntercept, it is most likely that many of these records from outside the Americas up to 2014 were simply misidentifications. Many of the Spodoptera that are intercepted are early instar, and difficult to identify to species. So it is not known why these particular interceptions were identified to the species level, and they were not verified with any other methods (sequencing, etc.). So it is possible they were identified correctly, but it is much more likely that most are simply misidentifications, and at this point we haven't gone back to sequence any of this material. So I think it would be good to add more ambiguity to the above statement if possible to let the reader know these IDs are not confirmed. Otherwise, I see no issues with the revised manuscript. Thank you for the opportunity to review!

[Authors]: We thank Reviewer #3 for the comments and suggested changes, and we have revised the statement (in red colour font below) to address the concern raised. The paragraph (P7L19-L26) now reads:

The pre-border interceptions of FAW **larvae (identified via morphological characters)** that originated from countries outside of the Americas since prior to 2014 ⁷¹ **(although molecular diagnostics of these suspect FAW larvae will be required to provide definitively confirmation of such non-native range interceptions)**, the early detections and report of FAW in Asia/S.E. Asia (e.g., CH06 ¹² (GenBank MT897262); 2008 Hanoi outbreaks ⁶⁹), and the complex pattern of multiple introductions including potential North American origins for various Chinese FAW populations (e.g., this study for individual CH06; the Yunnan 'NJ05' Individual ⁷²), are consistent with the perceived rapid spread experienced across the African ⁷³ and Asian continents ⁷⁴.

Communications Biology Editorial Team's comment:

Additionally, we recommend the following specific edits:

1. On P8L29, please change "Hexp < Hobs" to "Hobs > Hexp" to help in clarifying the previous reviewer 2 comment

[Authors]: Done, please see P4L21

2. Please make sure the CSIRO data is accessible to the public prior to official acceptance, as it is currently not,

[Authors]: Checked and confirmed that the data is accessible. The data has a permanent link at: <https://doi.org/10.25919/y3nd-2903>

3. the "FAW" from the title should be replaced by "Spodoptera frugiperda (fall armyworm)".

[Authors]: We have accepted the suggested new title provided in the Final Revision Instructions document. The new title for the manuscript now reads:

Global population genomic signature of *Spodoptera frugiperda* (fall armyworm) supports complex introduction events across the Old World